# Solving Inverse Problems using Diffusion with Iterative Colored Renoising

**Matthew C. Bendel**                                    *bendel.8@buckeyemail.osu.edu*
*Dept. Electrical and Computer Engineering, The Ohio State University, Columbus, OH, USA*

**Saurav K. Shastri**                                    *shastri.19@buckeyemail.osu.edu*
*Dept. Electrical and Computer Engineering, The Ohio State University, Columbus, OH, USA*

**Rizwan Ahmad**                                    *rizwan.ahmad@osumc.edu*
*Dept. Biomedical Engineering, The Ohio State University, Columbus, OH, USA*

**Philip Schniter**                                    *schniter.1@osu.edu*
*Dept. Electrical and Computer Engineering, The Ohio State University, Columbus, OH, USA*

**Reviewed on OpenReview:** *https://openreview.net/forum?id=RZv8FcQDPW*

## Abstract

Imaging inverse problems can be solved in an unsupervised manner using pre-trained diffusion models, but doing so requires approximating the gradient of the measurement-conditional score function in the diffusion reverse process. We show that the approximations produced by existing methods are relatively poor, especially early in the reverse process, and so we propose a new approach that iteratively reestimates and "renoises" the estimate several times per diffusion step. This iterative approach, which we call Fast Iterative REnoising (FIRE), injects colored noise that is shaped to ensure that the pre-trained diffusion model always sees white noise, in accordance with how it was trained. We then embed FIRE into the DDIM reverse process and show that the resulting "DDfire" offers state-of-the-art accuracy and runtime on several linear inverse problems, as well as phase retrieval. Our implementation is available at `https://github.com/matt-bendel/DDfire`.

## 1 Introduction

Diffusion has emerged as a powerful approach to generate samples from a complex distribution $p_0$ (Sohl-Dickstein et al., 2015; Song & Ermon, 2019; Ho et al., 2020; Song et al., 2021b;a). Recently, diffusion has also been used to solve inverse problems (Daras et al., 2024), where the goal is to recover $\boldsymbol{x}_0 \sim p_0$ from incomplete, distorted, and/or noisy measurements $\boldsymbol{y}$ in an unsupervised manner. There, a diffusion model is trained to generate samples from $p_0$ and, at test time, the reverse process is modified to incorporate knowledge of the measurements $\boldsymbol{y}$, with the goal of sampling from the posterior distribution $p(\boldsymbol{x}_0|\boldsymbol{y})$.

When implementing the reverse process, the main challenge is approximating the conditional score function $\nabla_{\boldsymbol{x}} \ln p_t(\boldsymbol{x}_t|\boldsymbol{y})$ at each step $t$, where $\boldsymbol{x}_t$ is an additive-white-Gaussian-noise (AWGN) corrupted and scaled version of $\boldsymbol{x}_0 \in \mathbb{R}^d$, and $\boldsymbol{y} \in \mathbb{R}^m$ is treated as a draw from a likelihood function $p(\boldsymbol{y}|\boldsymbol{x}_0)$. (See Sec. 2 for more details.) Many existing approaches fall into one of two categories. The first uses Bayes rule to write $\nabla_{\boldsymbol{x}} \ln p_t(\boldsymbol{x}_t|\boldsymbol{y}) = \nabla_{\boldsymbol{x}} \ln p_t(\boldsymbol{x}_t) + \nabla_{\boldsymbol{x}} \ln p_t(\boldsymbol{y}|\boldsymbol{x}_t)$, where an approximation of $\nabla_{\boldsymbol{x}} \ln p_t(\boldsymbol{x}_t)$ is readily available from the $p_0$-trained diffusion model, and then approximates $\nabla_{\boldsymbol{x}} \ln p_t(\boldsymbol{y}|\boldsymbol{x}_t)$ (e.g., Chung et al. (2023a); Song

et al. (2023)). The second approach uses Tweedie's formula (Efron, 2011) to write

$$\nabla_{\boldsymbol{x}} \ln p_t(\boldsymbol{x}_t|\boldsymbol{y}) = \frac{\mathrm{E}\{\boldsymbol{x}_0|\boldsymbol{x}_t, \boldsymbol{y}\} - \boldsymbol{x}_t}{\sigma_t^2} \tag{1}$$

and then approximates the conditional denoiser $\mathrm{E}\{\boldsymbol{x}_0|\boldsymbol{x}_t, \boldsymbol{y}\}$ (e.g., Kawar et al. (2022a); Wang et al. (2023); Zhu et al. (2023); Chung et al. (2024a)).

A key shortcoming of the aforementioned approaches is that their conditional-score approximations are not very accurate, especially early in the reverse process. For the methods that approximate $\mathrm{E}\{\boldsymbol{x}_0|\boldsymbol{x}_t, \boldsymbol{y}\}$, we can assess the approximation quality both visually and via mean-square error (MSE) or PSNR, since the exact $\mathrm{E}\{\boldsymbol{x}_0|\boldsymbol{x}_t, \boldsymbol{y}\}$ minimizes MSE given $\boldsymbol{x}_t$ and $\boldsymbol{y}$. For the methods that approximate $\nabla_{\boldsymbol{x}} \ln p_t(\boldsymbol{x}_t|\boldsymbol{y})$, we can compute their equivalent conditional-denoiser approximations using

$$\mathrm{E}\{\boldsymbol{x}_0|\boldsymbol{x}_t, \boldsymbol{y}\} = \boldsymbol{x}_t + \sigma_t^2 \nabla_{\boldsymbol{x}} \ln p_t(\boldsymbol{x}_t|\boldsymbol{y}), \tag{2}$$

which follows from (1). Figure 1 shows $\mathrm{E}\{\boldsymbol{x}_0|\boldsymbol{x}_t, \boldsymbol{y}\}$-approximations from the DDRM (Kawar et al., 2022a), DiffPIR (Zhu et al., 2023), DPS (Chung et al., 2023a), and DAPS Zhang et al. (2025) solvers at times 25%, 50%, and 75% through their reverse processes for noisy box inpainting with $\sigma_{\mathsf{y}} = 0.05$. The approximations show unwanted artifacts, especially early in the reverse process.

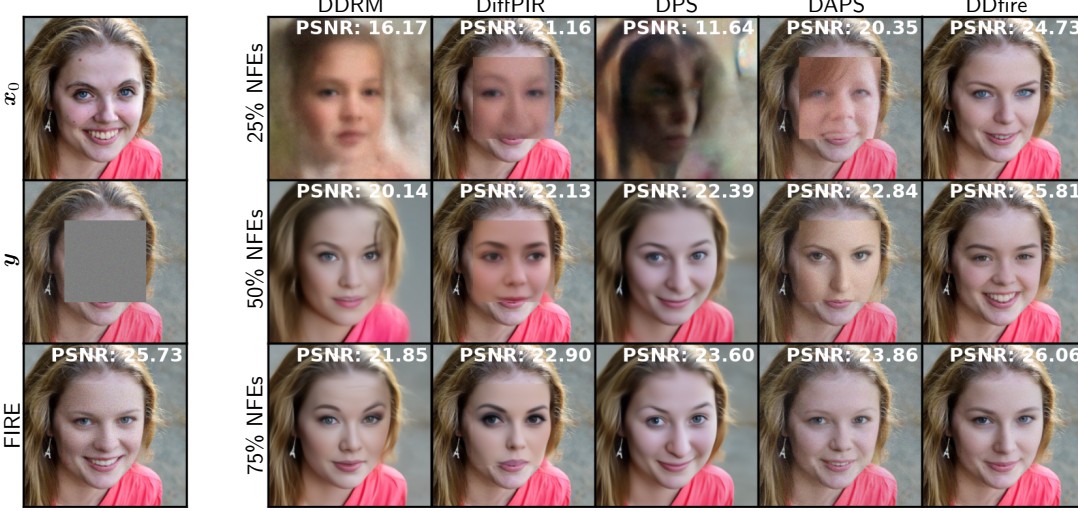

Figure 1: Left column: True $\boldsymbol{x}_0$, noisy box inpainting $\boldsymbol{y}$, and 50-iteration FIRE approximation of $\mathrm{E}\{\boldsymbol{x}_0|\boldsymbol{y}\}$. Other columns: Approximations of $\mathrm{E}\{\boldsymbol{x}_0|\boldsymbol{x}_t, \boldsymbol{y}\}$ at different $t$ (as measured by % NFEs). Note the over-smoothing with DDRM and DPS. Additionally, note the cut-and-paste artifacts of DiffPIR and DAPS.

In this paper, we aim to improve the approximation of $\mathrm{E}\{\boldsymbol{x}_0|\boldsymbol{x}_t, \boldsymbol{y}\}$ at each step $t$. Since we have observed that methods using a single neural function evaluation (NFE) to approximate $\mathrm{E}\{\boldsymbol{x}_0|\boldsymbol{x}_t, \boldsymbol{y}\}$ or $\nabla_{\boldsymbol{x}} \ln p_t(\boldsymbol{x}_t|\boldsymbol{y})$ perform poorly, we consider using several NFEs. In particular, we propose an iterative approach to approximating $\mathrm{E}\{\boldsymbol{x}_0|\boldsymbol{x}_t, \boldsymbol{y}\}$ that we call Fast Iterative REnoising (FIRE). FIRE is like a plug-and-play (PnP) algorithm (see the PnP survey Ahmad et al. (2020)) in that it iterates unconditional denoising with linear estimation from $\boldsymbol{x}_t$ and $\boldsymbol{y}$. But unlike traditional PnP algorithms, which aim to minimize an implicit loss function, FIRE is based on colored renoising, where carefully designed colored Gaussian noise is added to the linear-estimation output so that the denoiser's input error approximates AWGN. Since the denoiser is trained to remove AWGN, renoising aims to minimize the distribution shift experienced during inference. Figure 1 shows the 50-iteration FIRE approximation to $\mathrm{E}\{\boldsymbol{x}_0|\boldsymbol{x}_T, \boldsymbol{y}\} = \mathrm{E}\{\boldsymbol{x}_0|\boldsymbol{y}\}$ for noisy box inpainting, which is relatively free of artifacts. We propose two versions of FIRE: one that handles AWGN-corrupted linear measurement models like (6), and one that handles generalized-linear inverse problems such as phase retrieval (Shechtman et al., 2015), Poisson regression (Figueiredo & Bioucas-Dias, 2010), and dequantization (Zymnis et al., 2010). The latter is based on expectation propagation (EP) (Minka, 2001; Bishop, 2007).

We then embed FIRE into the DDIM diffusion reverse process (Song et al., 2021a), yielding the "DDfire" posterior sampler. Figure 1 shows examples of DDfire's $\mathrm{E}\{\boldsymbol{x}_0|\boldsymbol{x}_t, \boldsymbol{y}\}$ approximation when run on noisy box inpainting. DDfire's $\mathrm{E}\{\boldsymbol{x}_0|\boldsymbol{x}_t, \boldsymbol{y}\}$ approximations have fewer structural artifacts and higher PSNR than its competitors.

The contributions of this work are as follows:

1. We propose linear FIRE, an iterative approach to solving AWGN-corrupted linear inverse problems that injects carefully designed colored Gaussian noise in order to whiten the denoiser's input error.

2. We theoretically analyze the convergence of linear FIRE.

3. We use expectation propagation (EP) to extend linear FIRE to generalized-linear inverse problems.

4. Combining FIRE and DDIM, we propose the DDfire diffusion posterior sampler.

5. We demonstrate the excellent accuracy of DDfire on several imaging inverse problems: box inpainting, Gaussian and motion blur, super-resolution, and phase retrieval.

## 2 Background

Given training data drawn from distribution $p_0$, diffusion models corrupt the data with ever-increasing amounts of noise and then learn to reverse that process in a way that can generate new samples from $p_0$. In the main text, we assume the variance-exploding (VE) diffusion formulation (Song et al., 2021b), whereas Appendix A provides details on the variance-preserving (VP) formulation, including DDPM (Ho et al., 2020) and DDIM (Song et al., 2021a).

The VE diffusion forward process can be written as a stochastic differential equation (SDE) $\mathrm{d}\boldsymbol{x} = \sqrt{\mathrm{d}[\sigma^2(t)]/\mathrm{d}t}\,\mathrm{d}\boldsymbol{w}$ over $t$ from 0 to $T$, where $\sigma^2(t)$ is a variance schedule and $\mathrm{d}\boldsymbol{w}$ is the standard Wiener process (SWP) (Song et al., 2021b). The corresponding reverse process runs the SDE $\mathrm{d}\boldsymbol{x} = -\sigma^2(t)\nabla_{\boldsymbol{x}} \ln p_t(\boldsymbol{x})\,\mathrm{d}t + \sqrt{\mathrm{d}[\sigma^2(t)]/\mathrm{d}t}\,\mathrm{d}\overline{\boldsymbol{w}}$ backwards over $t$ from $T$ to 0, where $p_t(\cdot)$ is the marginal distribution of $\boldsymbol{x}$ at $t$ and $\mathrm{d}\overline{\boldsymbol{w}}$ is the SWP run backwards. The "score function" $\nabla_{\boldsymbol{x}} \ln p_t(\boldsymbol{x})$ can be approximated using a deep neural network (DNN) $\boldsymbol{s_\theta}(\boldsymbol{x}, t)$ trained via denoising score matching (Hyvärinen, 2005).

In practice, time is discretized to $t \in \{0, 1, \ldots, T\}$, yielding the SMLD from Song & Ermon (2019), whose forward process, $\boldsymbol{x}_{t+1} = \boldsymbol{x}_t + \sqrt{\sigma_{t+1}^2 - \sigma_t^2}\boldsymbol{w}_t$, with i.i.d $\{\boldsymbol{w}_t\} \sim \mathcal{N}(\boldsymbol{0}, \boldsymbol{I})$ and $\sigma_0^2 = 0$, implies that

$$\boldsymbol{x}_t = \boldsymbol{x}_0 + \sigma_t \boldsymbol{\epsilon}_t, \quad \boldsymbol{\epsilon}_t \sim \mathcal{N}(\boldsymbol{0}, \boldsymbol{I}) \tag{3}$$

for all $t \in \{0, 1, \ldots, T\}$. The SMLD reverse process then uses i.i.d $\{\boldsymbol{n}_t\} \sim \mathcal{N}(\boldsymbol{0}, \boldsymbol{I})$ in

$$\boldsymbol{x}_t = \boldsymbol{x}_{t+1} + (\sigma_{t+1}^2 - \sigma_t^2)\nabla_{\boldsymbol{x}} \ln p_{t+1}(\boldsymbol{x}_{t+1}) + \sqrt{\frac{\sigma_t^2(\sigma_{t+1}^2 - \sigma_t^2)}{\sigma_{t+1}^2}}\boldsymbol{n}_{t+1}. \tag{4}$$

To exploit side information about $\boldsymbol{x}_0$, such as the measurements $\boldsymbol{y}$ in an inverse problem, one can simply replace $p_t(\cdot)$ with $p_t(\cdot|\boldsymbol{y})$ in the above equations (Song et al., 2021b). However, most works aim to avoid training a $\boldsymbol{y}$-dependent approximation of the conditional score function $\nabla_{\boldsymbol{x}} \ln p_t(\boldsymbol{x}_t|\boldsymbol{y})$. Rather, they take an "unsupervised" approach, where $\boldsymbol{s_\theta}(\boldsymbol{x}_t, t) \approx \nabla_{\boldsymbol{x}} \ln p_t(\boldsymbol{x}_t)$ is learned during training but $\boldsymbol{y}$ is presented only at inference (Daras et al., 2024). In this case, approximating $\nabla_{\boldsymbol{x}} \ln p_t(\boldsymbol{x}_t|\boldsymbol{y})$ is the key technical challenge.

There are two major approaches to approximate $\nabla_{\boldsymbol{x}} \ln p_t(\boldsymbol{x}_t|\boldsymbol{y})$. The first uses the Bayes rule to write $\nabla_{\boldsymbol{x}} \ln p_t(\boldsymbol{x}_t|\boldsymbol{y}) = \nabla_{\boldsymbol{x}} \ln p_t(\boldsymbol{x}_t) + \nabla_{\boldsymbol{x}} \ln p_t(\boldsymbol{y}|\boldsymbol{x}_t)$ and then replaces $\nabla_{\boldsymbol{x}} \ln p_t(\boldsymbol{x}_t)$ with the score approximation $\boldsymbol{s_\theta}(\boldsymbol{x}_t, t)$. But the remaining term, $\nabla_{\boldsymbol{x}} \ln p_t(\boldsymbol{y}|\boldsymbol{x}_t)$, is intractable because $p_t(\boldsymbol{y}|\boldsymbol{x}_t) = \int p(\boldsymbol{y}|\boldsymbol{x}_0)p(\boldsymbol{x}_0|\boldsymbol{x}_t)\,\mathrm{d}\boldsymbol{x}_0$ with unknown $p(\boldsymbol{x}_0|\boldsymbol{x}_t)$, and so several approximations have been proposed. For example, DPS (Chung et al., 2023a) uses $p(\boldsymbol{x}_0|\boldsymbol{x}_t) \approx \delta(\boldsymbol{x}_0 - \widehat{\boldsymbol{x}}_{0|t})$, where $\widehat{\boldsymbol{x}}_{0|t}$ is the approximation of $\mathrm{E}\{\boldsymbol{x}_0|\boldsymbol{x}_t\}$ computed from $\boldsymbol{s_\theta}(\boldsymbol{x}_t, t)$ using Tweedie's formula:

$$\widehat{\boldsymbol{x}}_{0|t} = \boldsymbol{x}_t + \sigma_t^2 \boldsymbol{s_\theta}(\boldsymbol{x}_t, t). \tag{5}$$

Similarly, ΠGDM (Song et al., 2023) uses $p(\boldsymbol{x}_0|\boldsymbol{x}_t) \approx \mathcal{N}(\boldsymbol{x}_0; \widehat{\boldsymbol{x}}_{0|t}, \zeta_t \boldsymbol{I})$ with some $\zeta_t$. However, a drawback to both approaches is that they require backpropagation through $\boldsymbol{s}_{\boldsymbol{\theta}}(\cdot, t)$, which increases the cost of generating a single sample. In Fig. 4, we show that DDfire offers a $1.5\times$ speedup over DPS at an equal number of NFEs.

The second major approach to approximating $\nabla_{\boldsymbol{x}} \ln p_t(\boldsymbol{x}_t|\boldsymbol{y})$ uses (1) with $\mathrm{E}\{\boldsymbol{x}_0|\boldsymbol{x}_t, \boldsymbol{y}\}$ approximated by a quantity that we'll refer to as $\widehat{\boldsymbol{x}}_{0|t,\boldsymbol{y}}$. For example, with AWGN-corrupted linear measurements

$$\boldsymbol{y} = \boldsymbol{A}\boldsymbol{x}_0 + \sigma_{\mathsf{y}}\boldsymbol{w} \in \mathbb{R}^m, \quad \boldsymbol{w} \sim \mathcal{N}(\boldsymbol{0}, \boldsymbol{I}), \tag{6}$$

DDNM (Wang et al., 2023) approximates $\mathrm{E}\{\boldsymbol{x}_0|\boldsymbol{x}_t, \boldsymbol{y}\}$ by first computing $\widehat{\boldsymbol{x}}_{0|t}$ from (5) and then performing the hard data-consistency step $\widehat{\boldsymbol{x}}_{0|t,\boldsymbol{y}} = \boldsymbol{A}^+\boldsymbol{y} + (\boldsymbol{I} - \boldsymbol{A}^+\boldsymbol{A})\widehat{\boldsymbol{x}}_{0|t}$, where $(\cdot)^+$ is the pseudo-inverse. DDS (Chung et al., 2024a) and DiffPIR (Zhu et al., 2023) instead use the soft data-consistency step $\widehat{\boldsymbol{x}}_{0|t,\boldsymbol{y}} = \arg\min_{\boldsymbol{x}} \|\boldsymbol{y} - \boldsymbol{A}\boldsymbol{x}\|^2 + \gamma_t \|\boldsymbol{x} - \widehat{\boldsymbol{x}}_{0|t}\|^2$ with some $\gamma_t > 0$. DDRM (Kawar et al., 2022a) is a related technique that requires a singular value decomposition (SVD), which is prohibitive in many applications.

There are other ways to design posterior samplers. For example, Laumont et al. (2022) and Jalal et al. (2021) use Langevin dynamics. RED-diff (Mardani et al., 2024) and SNORE (Renaud et al., 2024) inject white noise into the RED algorithm (Romano et al., 2017), whose regularizer's gradient equals the score function (Reehorst & Schniter, 2019). Bouman & Buzzard (2023); Coeurdoux et al. (2024); Wu et al. (2024); Xu & Chi (2024); Zhang et al. (2025) use Markov-chain Monte Carlo (MCMC) in the diffusion reverse process.

## 3 Approach

We aim to accurately approximate the conditional denoiser $\mathrm{E}\{\boldsymbol{x}_0|\boldsymbol{x}_t, \boldsymbol{y}\}$ at each step of the diffusion reverse process. We treat the standard linear model (SLM) (6) first and the generalized linear model (GLM) later.

### 3.1 Fast iterative renoising (FIRE) for the SLM

In this section, we describe the FIRE algorithm, which approximates $\mathrm{E}\{\boldsymbol{x}_0|\boldsymbol{r}_{\mathsf{init}}, \boldsymbol{y}\}$ assuming $\boldsymbol{y}$ from (6) and $\boldsymbol{r}_{\mathsf{init}} = \boldsymbol{x}_0 + \sigma_{\mathsf{init}}\boldsymbol{\epsilon}$ with $\boldsymbol{\epsilon} \sim \mathcal{N}(\boldsymbol{0}, \boldsymbol{I})$ and some $\sigma_{\mathsf{init}} > 0$. FIRE performs half-quadratic splitting (HQS) PnP with a scheduled denoising variance $\sigma^2$, similar to DPIR from Zhang et al. (2021), but injects colored noise $\boldsymbol{c}$ to ensure that the error in the denoiser input $\boldsymbol{r}$ remains white. It is beneficial for the denoiser to see white input error during inference, because it is trained to remove white input error. The basic FIRE algorithm iterates the following steps $N \geq 1$ times, after initializing $\boldsymbol{r} \leftarrow \boldsymbol{r}_{\mathsf{init}}$ and $\sigma \leftarrow \sigma_{\mathsf{init}}$:

    S1) Denoise $\boldsymbol{r}$ assuming AWGN of variance $\sigma^2$, giving $\overline{\boldsymbol{x}}$.

    S2) MMSE estimate $\boldsymbol{x}_0$ given $\boldsymbol{y}$ from (6) and the prior $\boldsymbol{x}_0 \sim \mathcal{N}(\overline{\boldsymbol{x}}, \nu\boldsymbol{I})$ with some $\nu > 0$, giving $\widehat{\boldsymbol{x}}$.

    S3) Update the denoising variance $\sigma^2$ via $\sigma^2 \leftarrow \sigma^2/\rho$ with some $\rho > 1$,

    S4) Update $\boldsymbol{r} \leftarrow \widehat{\boldsymbol{x}} + \boldsymbol{c}$ using colored Gaussian noise $\boldsymbol{c}$ created to ensure $\mathrm{Cov}\{\boldsymbol{r} - \boldsymbol{x}_0\} = \sigma^2\boldsymbol{I}$.

S1)-S3) are essentially DPIR with regularization strength controlled by $\nu$ and a geometric denoising schedule with rate controlled by $\rho$, while S4) injects colored noise. In contrast, other renoising PnP approaches like SNORE (Renaud et al., 2024) inject white noise. Section 4.1 examines the effect of removing $\boldsymbol{c}$ or replacing it with white noise. Next we provide details and enhancements of the basic FIRE algorithm.

In the sequel, we use "$\boldsymbol{d}_{\boldsymbol{\theta}}(\boldsymbol{x}, \sigma)$" to denote a neural-net approximation of the conditional-mean denoiser $\mathrm{E}\{\boldsymbol{x}_0|\boldsymbol{x}\}$ of $\boldsymbol{x} = \boldsymbol{x}_0 + \sigma\boldsymbol{\epsilon}$ with $\boldsymbol{\epsilon} \sim \mathcal{N}(\boldsymbol{0}, \boldsymbol{I})$. Given a score function approximation $\boldsymbol{s}_{\boldsymbol{\theta}}(\boldsymbol{x}, t) \approx \nabla_{\boldsymbol{x}} \ln p_t(\boldsymbol{x})$ as discussed in Sec. 2, the denoiser can be constructed via (5) as

$$\boldsymbol{d}_{\boldsymbol{\theta}}(\boldsymbol{x}, \sigma) = \boldsymbol{x} + \sigma^2 \boldsymbol{s}_{\boldsymbol{\theta}}(\boldsymbol{x}, t) \text{ with } t \text{ such that } \sigma_t = \sigma. \tag{7}$$

When FIRE estimates $\boldsymbol{x}_0$ from the measurements $\boldsymbol{y}$ and the denoiser output $\overline{\boldsymbol{x}}$, it employs a Gaussian approximation of the form $\boldsymbol{x}_0 \sim \mathcal{N}(\overline{\boldsymbol{x}}, \nu\boldsymbol{I})$, similar to DDS, DiffPIR, and prox-based PnP algorithms. But it differs in that $\nu$ is explicitly estimated. The Gaussian approximation $\boldsymbol{x}_0 \sim \mathcal{N}(\overline{\boldsymbol{x}}, \nu\boldsymbol{I})$ is equivalent to

$$\boldsymbol{x}_0 = \overline{\boldsymbol{x}} + \sqrt{\nu}\boldsymbol{e}, \quad \boldsymbol{e} \sim \mathcal{N}(\boldsymbol{0}, \boldsymbol{I}). \tag{8}$$

Suppose $\boldsymbol{x}_0 = \overline{\boldsymbol{x}} + \sqrt{\nu_0}\boldsymbol{e}$ with $\boldsymbol{e} \sim \mathcal{N}(\boldsymbol{0}, \boldsymbol{I})$, where $\nu_0$ denotes the true error variance. Then (6) and (8) imply

$$\mathrm{E}\{\|\boldsymbol{y} - \boldsymbol{A}\overline{\boldsymbol{x}}\|^2\} = \mathrm{E}\{\|\boldsymbol{A}\boldsymbol{x}_0 + \sigma_{\mathsf{y}}\boldsymbol{w} - \boldsymbol{A}\boldsymbol{x}_0 + \sqrt{\nu_0}\boldsymbol{A}\boldsymbol{e}\|^2\} = \mathrm{E}\{\|\sigma_{\mathsf{y}}\boldsymbol{w} + \sqrt{\nu_0}\boldsymbol{A}\boldsymbol{e}\|^2\} = m\sigma_{\mathsf{y}}^2 + \nu_0\|\boldsymbol{A}\|_F^2, \quad (9)$$

assuming independence between $\boldsymbol{e}$ and $\boldsymbol{w}$. Consequently, an unbiased estimate of $\nu_0$ can be constructed as

$$\left(\|\boldsymbol{y} - \boldsymbol{A}\overline{\boldsymbol{x}}\|^2 - m\sigma_{\mathsf{y}}^2\right)/\|\boldsymbol{A}\|_F^2 \triangleq \nu. \tag{10}$$

Figure 12 shows that, in practice, the estimate (10) accurately tracks the true error variance $\|\boldsymbol{x}_0 - \overline{\boldsymbol{x}}\|^2/d$. Under (6) and (8), the MMSE estimate of $\boldsymbol{x}_0$ from $\boldsymbol{y}$ and $\overline{\boldsymbol{x}}$ can be written as (Poor, 1994)

$$\widehat{\boldsymbol{x}} \triangleq \arg\min_{\boldsymbol{x}} \left\{ \frac{1}{2\sigma_{\mathsf{y}}^2}\|\boldsymbol{y} - \boldsymbol{A}\boldsymbol{x}\|^2 + \frac{1}{2\nu}\|\boldsymbol{x} - \overline{\boldsymbol{x}}\|^2 \right\} = \left(\boldsymbol{A}^\mathsf{T}\boldsymbol{A} + \frac{\sigma_{\mathsf{y}}^2}{\nu}\boldsymbol{I}\right)^{-1}\left(\boldsymbol{A}^\mathsf{T}\boldsymbol{y} + \frac{\sigma_{\mathsf{y}}^2}{\nu}\overline{\boldsymbol{x}}\right). \tag{11}$$

Equation (11) can be computed using conjugate gradients (CG) or, if practical, the SVD $\boldsymbol{A} = \boldsymbol{U}\boldsymbol{S}\boldsymbol{V}^\mathsf{T}$ via

$$\widehat{\boldsymbol{x}} = \boldsymbol{V}\left(\boldsymbol{S}^\mathsf{T}\boldsymbol{S} + \frac{\sigma_{\mathsf{y}}^2}{\nu}\boldsymbol{I}\right)^{-1}\left(\boldsymbol{S}^\mathsf{T}\boldsymbol{U}^\mathsf{T}\boldsymbol{y} + \frac{\sigma_{\mathsf{y}}^2}{\nu}\boldsymbol{V}^\mathsf{T}\overline{\boldsymbol{x}}\right). \tag{12}$$

In any case, from (8) and (11), the error in $\widehat{\boldsymbol{x}}$ can be written as

$$\widehat{\boldsymbol{x}} - \boldsymbol{x}_0 = \left(\boldsymbol{A}^\mathsf{T}\boldsymbol{A} + \frac{\sigma_{\mathsf{y}}^2}{\nu}\boldsymbol{I}\right)^{-1}\left(\boldsymbol{A}^\mathsf{T}[\boldsymbol{A}\boldsymbol{x}_0 + \sigma_{\mathsf{y}}\boldsymbol{w}] + \frac{\sigma_{\mathsf{y}}^2}{\nu}[\boldsymbol{x}_0 - \sqrt{\nu}\boldsymbol{e}]\right) - \boldsymbol{x}_0 = \left(\boldsymbol{A}^\mathsf{T}\boldsymbol{A} + \frac{\sigma_{\mathsf{y}}^2}{\nu}\boldsymbol{I}\right)^{-1}\left(\sigma_{\mathsf{y}}\boldsymbol{A}^\mathsf{T}\boldsymbol{w} - \frac{\sigma_{\mathsf{y}}^2}{\sqrt{\nu}}\boldsymbol{e}\right), \tag{13}$$

and so the covariance of the error in $\widehat{\boldsymbol{x}}$ can be written as

$$\mathrm{Cov}\{\widehat{\boldsymbol{x}} - \boldsymbol{x}_0\} = \left(\boldsymbol{A}^\mathsf{T}\boldsymbol{A} + \frac{\sigma_{\mathsf{y}}^2}{\nu}\boldsymbol{I}\right)^{-1}\left(\sigma_{\mathsf{y}}^2\boldsymbol{A}^\mathsf{T}\boldsymbol{A} + \frac{\sigma_{\mathsf{y}}^4}{\nu}\boldsymbol{I}\right)\left(\boldsymbol{A}^\mathsf{T}\boldsymbol{A} + \frac{\sigma_{\mathsf{y}}^2}{\nu}\boldsymbol{I}\right)^{-1} = \left(\frac{1}{\sigma_{\mathsf{y}}^2}\boldsymbol{A}^\mathsf{T}\boldsymbol{A} + \frac{1}{\nu}\boldsymbol{I}\right)^{-1} \triangleq \boldsymbol{C}. \tag{14}$$

From (14) we see that the error in $\widehat{\boldsymbol{x}}$ can be strongly colored. For example, in the case of inpainting, where $\boldsymbol{A}$ is formed from rows of the identity matrix, the error variance in the masked pixels equals $\nu$, while the error variance in the unmasked pixels equals $(1/\sigma_{\mathsf{y}}^2 + 1/\nu)^{-1} \leq \sigma_{\mathsf{y}}^2$. These two values may differ by many orders of magnitude. Since most denoisers are trained to remove white noise with a specified variance of $\sigma^2$, direct denoising of $\widehat{\boldsymbol{x}}$ performs poorly, as we show in Sec. 4.1.

To circumvent the issues that arise from colored denoiser-input error, we propose to add "complementary" colored Gaussian noise $\boldsymbol{c} \sim \mathcal{N}(\boldsymbol{0}, \boldsymbol{\Sigma})$ to $\widehat{\boldsymbol{x}}$ so that the resulting $\boldsymbol{r} = \widehat{\boldsymbol{x}} + \boldsymbol{c}$ has an error covariance of $\sigma^2\boldsymbol{I}$, i.e., white error. This requires that

$$\boldsymbol{\Sigma} = \sigma^2\boldsymbol{I} - \boldsymbol{C} = \sigma^2\boldsymbol{I} - \left(\frac{1}{\sigma_{\mathsf{y}}^2}\boldsymbol{V}\boldsymbol{S}^\mathsf{T}\boldsymbol{S}\boldsymbol{V}^\mathsf{T} + \frac{1}{\nu}\boldsymbol{I}\right)^{-1} = \boldsymbol{V}\,\mathrm{Diag}(\boldsymbol{\lambda})\boldsymbol{V}^\mathsf{T} \text{ for } \lambda_i = \sigma^2 - \frac{1}{s_i^2/\sigma_{\mathsf{y}}^2 + 1/\nu} \tag{15}$$

for $s_i^2 \triangleq [\boldsymbol{S}^\mathsf{T}\boldsymbol{S}]_{i,i}$. By setting $\sigma^2 \geq \nu$, we ensure that $\lambda_i \geq 0 \ \forall i$, needed for $\boldsymbol{\Sigma}$ to be a valid covariance matrix. In the case that the SVD is practical to implement, we can generate $\boldsymbol{c}$ using

$$\boldsymbol{c} = \boldsymbol{V}\,\mathrm{Diag}(\boldsymbol{\lambda})^{1/2}\boldsymbol{\varepsilon}, \quad \boldsymbol{\varepsilon} \sim \mathcal{N}(\boldsymbol{0}, \boldsymbol{I}). \tag{16}$$

In the absence of an SVD, we propose to approximate $\boldsymbol{\Sigma}$ by

$$\widehat{\boldsymbol{\Sigma}} \triangleq (\sigma^2 - \nu)\boldsymbol{I} + \xi\boldsymbol{A}^\mathsf{T}\boldsymbol{A} \tag{17}$$

with some $\xi \geq 0$. Note that $\widehat{\boldsymbol{\Sigma}}$ agrees with $\boldsymbol{\Sigma}$ in the nullspace of $\boldsymbol{A}$ (i.e., when $s_n = 0$) for any $\xi$. By choosing

$$\xi = \frac{1}{s_{\max}^2}\left(\nu - \frac{1}{s_{\max}^2/\sigma_{\mathsf{y}}^2 + 1/\nu}\right), \tag{18}$$

---

**Algorithm 1** FIRE for the SLM: $\widehat{\boldsymbol{x}} = \mathsf{FIRE}_{\mathsf{SLM}}(\boldsymbol{y}, \boldsymbol{A}, \sigma_{\mathsf{y}}, \boldsymbol{r}_{\mathsf{init}}, \sigma_{\mathsf{init}}, N, \rho)$

---

**Require:** $\boldsymbol{d}_{\boldsymbol{\theta}}(\cdot, \cdot), \boldsymbol{y}, \boldsymbol{A}, s_{\mathsf{max}}, \sigma_{\mathsf{y}}, N, \rho > 1, \boldsymbol{r}_{\mathsf{init}}, \sigma_{\mathsf{init}}$. Also $\boldsymbol{A} = \boldsymbol{U} \operatorname{Diag}(\boldsymbol{s}) \boldsymbol{V}^{\mathsf{T}}$ if using SVD.

1: $\boldsymbol{r} = \boldsymbol{r}_{\mathsf{init}}$ and $\sigma = \sigma_{\mathsf{init}}$ ▷ *Initialize*
2: **for** $n = 1, \ldots, N$ **do**
3:     $\overline{\boldsymbol{x}} \leftarrow \boldsymbol{d}_{\boldsymbol{\theta}}(\boldsymbol{r}, \sigma) + \sqrt{\widehat{\nu}_{\boldsymbol{\phi}}(\sigma)}\boldsymbol{v}, \;\; \boldsymbol{v} \sim \mathcal{N}(\boldsymbol{0}, \boldsymbol{I})$ ▷ *Stochastic denoising*
4:     $\nu \leftarrow (\|\boldsymbol{y} - \boldsymbol{A}\overline{\boldsymbol{x}}\|^2 - \sigma_{\mathsf{y}}^2 m) / \|\boldsymbol{A}\|_F^2$ ▷ *Error variance of $\overline{\boldsymbol{x}}$*
5:     $\widehat{\boldsymbol{x}} \leftarrow \arg\min_{\boldsymbol{x}} \|\boldsymbol{y} - \boldsymbol{A}\boldsymbol{x}\|^2 / \sigma_{\mathsf{y}}^2 + \|\boldsymbol{x} - \overline{\boldsymbol{x}}\|^2 / \nu$ ▷ *Estimate $\boldsymbol{x}_0 \sim \mathcal{N}(\overline{\boldsymbol{x}}, \nu \boldsymbol{I})$ from $\boldsymbol{y} \sim \mathcal{N}(\boldsymbol{A}\boldsymbol{x}_0, \sigma_{\mathsf{y}}^2 \boldsymbol{I})$*
6:     $\sigma^2 \leftarrow \max\{\sigma^2/\rho, \nu\}$ ▷ *Decrease target variance*
7:     **if** have SVD **then**
8:        $\lambda_i \leftarrow \sigma^2 - (s_i^2/\sigma_{\mathsf{y}}^2 + 1/\nu)^{-1}, \;\; i = 1, \ldots, d$
9:        $\boldsymbol{c} \leftarrow \boldsymbol{V} \operatorname{Diag}(\boldsymbol{\lambda})^{1/2} \boldsymbol{\varepsilon}, \;\; \boldsymbol{\varepsilon} \sim \mathcal{N}(\boldsymbol{0}, \boldsymbol{I})$ ▷ *Colored Gaussian noise*
10:     **else**
11:        $\xi \leftarrow (\nu - (s_{\mathsf{max}}^2/\sigma_{\mathsf{y}}^2 + 1/\nu)^{-1})/s_{\mathsf{max}}^2$
12:        $\boldsymbol{c} \leftarrow [\sqrt{\sigma^2 - \nu}\boldsymbol{I} \;\; \sqrt{\xi}\boldsymbol{A}^{\mathsf{T}}]\boldsymbol{\varepsilon}, \;\; \boldsymbol{\varepsilon} \sim \mathcal{N}(\boldsymbol{0}, \boldsymbol{I})$ ▷ *Colored Gaussian noise*
13:     $\boldsymbol{r} \leftarrow \widehat{\boldsymbol{x}} + \boldsymbol{c}$ ▷ *Renoise so that $\operatorname{Cov}\{\boldsymbol{r} - \boldsymbol{x}_0\} = \sigma^2 \boldsymbol{I}$*
14: **return** $\widehat{\boldsymbol{x}}$

---

$\widehat{\boldsymbol{\Sigma}}$ will also agree with $\boldsymbol{\Sigma}$ in the strongest measured subspace (i.e., when $s_n = s_{\mathsf{max}}$). Without an SVD, $s_{\mathsf{max}}$ can be computed using the power iteration (Parlett, 1998). Finally, $\boldsymbol{c} \sim \mathcal{N}(\boldsymbol{0}, \widehat{\boldsymbol{\Sigma}})$ can be generated via

$$\boldsymbol{c} = [\sqrt{\sigma^2 - \nu}\boldsymbol{I} \;\; \sqrt{\xi}\boldsymbol{A}^{\mathsf{T}}]\boldsymbol{\varepsilon}, \;\; \boldsymbol{\varepsilon} \sim \mathcal{N}(\boldsymbol{0}, \boldsymbol{I}) \in \mathbb{R}^{d+m}. \tag{19}$$

Figure 6 shows a close agreement between the ideal and approximate renoised error spectra in practice. Next, we provide the main theoretical result on FIRE.

**Theorem 1** *Suppose that, for any input $\boldsymbol{r} = \boldsymbol{x}_0 + \sigma\boldsymbol{\epsilon}$ with $\boldsymbol{\epsilon} \sim \mathcal{N}(\boldsymbol{0}, \boldsymbol{I})$, the denoiser output $\boldsymbol{d}_{\boldsymbol{\theta}}(\boldsymbol{r}, \sigma)$ has white Gaussian error with known variance $\nu < \sigma^2$ and independent of the noise $\boldsymbol{w}$ in (6). Then if initialized using $\boldsymbol{r}_{\mathsf{init}} = \boldsymbol{x}_0 + \sigma_{\mathsf{init}}\boldsymbol{\epsilon}$ with arbitrarily large but finite $\sigma_{\mathsf{init}}$ and $\boldsymbol{\epsilon} \sim \mathcal{N}(\boldsymbol{0}, \boldsymbol{I})$, there exists a $\rho > 1$ under which the FIRE iteration S1)-S4) converges to the true $\boldsymbol{x}_0$.*

Appendix E provides a proof. Note that a key assumption of Theorem 1 is that the denoiser output error is white and Gaussian. Because this may not hold in practice, we propose to replace S1) with a "stochastic denoising" step (21), in which AWGN is explicitly added to the denoiser output. As the AWGN variance increases, the denoiser output becomes closer to white and Gaussian but its signal-to-noise ratio (SNR) degrades. To balance these competing objectives, we propose to add AWGN with variance approximately equal to that of the raw-denoiser output error. We estimate the latter quantity from the denoiser input variance $\sigma^2$ by training a predictor of the form

$$\widehat{\nu}_{\boldsymbol{\phi}}(\sigma) \approx \mathrm{E}\{\|\boldsymbol{d}_{\boldsymbol{\theta}}(\boldsymbol{x}_0 + \sigma\boldsymbol{\epsilon}, \sigma) - \boldsymbol{x}_0\|^2/d\}, \tag{20}$$

where the expectation is over $\boldsymbol{\epsilon} \sim \mathcal{N}(\boldsymbol{0}, \boldsymbol{I})$ and validation images $\boldsymbol{x}_0 \sim p_0$. Recall that $d$ is the dimension of $\boldsymbol{x}_0$. In our experiments, $\widehat{\nu}_{\boldsymbol{\phi}}(\cdot)$ is implemented using a lookup table. The stochastic denoising step is then

$$\overline{\boldsymbol{x}} = \boldsymbol{d}_{\boldsymbol{\theta}}(\boldsymbol{x}, \sigma) + \sqrt{\widehat{\nu}_{\boldsymbol{\phi}}(\sigma)}\boldsymbol{v}, \;\; \boldsymbol{v} \sim \mathcal{N}(\boldsymbol{0}, \boldsymbol{I}). \tag{21}$$

Algorithm 1 summarizes the FIRE algorithm for the SLM (6). In App. D, we describe a minor enhancement to Alg. 1 that speeds up the MMSE estimation step when CG is used.

### 3.2 DDfire for the GLM

We now propose to extend the SLM-FIRE from Sec. 3.1 to the generalized linear model (GLM)

$$\boldsymbol{y} \sim p(\boldsymbol{y}|\boldsymbol{z}_0) = \prod_{j=1}^{m} p_{\mathsf{y}|\mathsf{z}}(y_j|z_{0,j}) \;\; \text{with} \;\; \boldsymbol{z}_0 \triangleq \boldsymbol{A}\boldsymbol{x}_0 \tag{22}$$

Figure 2: High-level overview of GLM-FIRE, which uses EP-style iterations between SLM-FIRE and an MMSE inference stage that involves the scalar measurement channel $p_{y|z}$.

where $p_{y|z}$ is some scalar "measurement channel." Examples include $p_{y|z}(y|z) = \mathcal{N}(y; |z|, \sigma_y^2)$ for phase retrieval, $p_{y|z}(y|z) = z^y e^{-z}/y!$ for Poisson regression, and $p_{y|z}(y|z) = \int_{\tau_y}^{\tau_{y+1}} \mathcal{N}(\tau; z, \sigma_y^2) \, d\tau$ for dequantization.

Our extension is inspired by expectation propagation (EP) (Minka, 2001; Bishop, 2007) and its application to GLMs (Schniter et al., 2016; Meng et al., 2018). The idea is to iterate between i) constructing "pseudo-measurements" $\overline{y} = A x_0 + \overline{w}$ with $\overline{w} \sim \mathcal{N}(0, \overline{\sigma}_y^2 I)$ using $p_{y|z}$ and an SLM-FIRE-constructed belief that $z_0 \sim \mathcal{N}(\overline{z}_0, \overline{\nu}_z I)$, and then ii) running SLM-FIRE with those pseudo-measurements and updating its belief on $z_0$. Figure 2 shows a high-level summary. Details are given below.

To construct the belief on $z_0$, we use the SLM-FIRE denoiser output model $x_0 \sim \mathcal{N}(\overline{x}, \nu I)$ from (8). Because $z_0 = A x_0$, we see that $z_0 \sim \mathcal{N}(\overline{z}, \nu A A^\mathsf{T})$, where $\overline{z} \triangleq A \overline{x}$. For simplicity, however, we use the white-noise approximation $z_0 \sim \mathcal{N}(\overline{z}, \overline{\nu}_z I)$, where $\overline{\nu}_z \triangleq \nu \|A\|_F^2 / m$. Using the scalar belief $z_{0,j} \sim \mathcal{N}(\overline{z}_j, \overline{\nu}_z)$ and the likelihood model $y_j \sim p_{y|z}(\cdot | z_{0,j})$, EP suggests to first compute the posterior mean $\mathrm{E}\{z_{0,j} | y_j; \overline{z}_j, \overline{\nu}_z\} \triangleq \widehat{z}_j$ and variance $\frac{1}{m} \sum_{j=1}^m \mathrm{var}\{z_{0,j} | y_j; \overline{z}_j, \overline{\nu}_z\} \triangleq \widehat{\nu}_z$, and then pass the "extrinsic" versions of those quantities:

$$\overline{\sigma}_y^2 \triangleq [1/\widehat{\nu}_z - 1/\overline{\nu}_z]^{-1}, \quad \overline{y} \triangleq \overline{\sigma}_y^2 (\widehat{z}/\widehat{\nu}_z - \overline{z}/\overline{\nu}_z) \tag{23}$$

back to SLM-FIRE, where they are used to construct the pseudo-measurement model

$$\overline{y} = A x_0 + \overline{\sigma}_y \overline{w}, \quad \overline{w} \sim \mathcal{N}(0, I). \tag{24}$$

In Fig. 13, we show that GLM-FIRE's $\overline{\sigma}_y^2$ accurately tracks the true noise power in $\overline{y}$, i.e., $\|\overline{y} - A x_0\|^2 / m$.

The GLM-FIRE algorithm is summarized as Alg. 3. When $p_{y|z}(y|z) = \mathcal{N}(y; z, \sigma_y^2)$, is it straightforward to show that $\overline{y} = y$ and $\overline{\sigma}_y^2 = \sigma_y^2$ for any $\overline{z}$ and $\overline{\nu}_z$, in which case GLM-FIRE reduces to SLM-FIRE.

## 3.3 Putting FIRE into diffusion

Sections 3.1 and 3.2 detailed the FIRE algorithms for the SLM (6) and the GLM (22), respectively. In both cases, the FIRE algorithm approximates $\mathrm{E}\{x_0 | r, y\}$ given the measurements $y$ and the side-information $r = x_0 + \sigma \epsilon$, where $\epsilon \sim \mathcal{N}(0, I)$. Thus, recalling the discussion in Sec. 2, FIRE can be used in the SMLD reverse process as an approximation of $\mathrm{E}\{x_0 | x_t, y\}$ by setting $r = x_t$ and $\sigma = \sigma_t$.

Instead of using SMLD for the diffusion reverse process, however, we use DDIM from Song et al. (2021a), which can be considered as a generalization of SMLD. In the sequel, we distinguish the DDIM quantities by writing them with subscript $k$. As detailed in App. C, DDIM is based on the model

$$x_k = x_0 + \sigma_k \epsilon_k, \quad \epsilon_k \sim \mathcal{N}(0, I), \tag{25}$$

for $k = 1, \ldots, K$, where $\{\sigma_k^2\}_{k=1}^K$ is a specified sequence of variances. The DDIM reverse process iterates

$$x_{k-1} = h_k x_k + g_k \mathrm{E}\{x_0 | x_k, y\} + \varsigma_k n_k \tag{26}$$

$$\varsigma_k = \eta_{\mathsf{ddim}} \sqrt{\frac{\sigma_{k-1}^2 (\sigma_k^2 - \sigma_{k-1}^2)}{\sigma_k^2}}, \quad h_k = \sqrt{\frac{\sigma_{k-1}^2 - \varsigma_k^2}{\sigma_k^2}}, \quad g_k = 1 - h_k \tag{27}$$

over $k = K, \ldots, 2, 1$, starting from $x_K \sim \mathcal{N}(0, \sigma_K^2 I)$, using i.i.d $\{n_k\}_{k=1}^K \sim \mathcal{N}(0, I)$ and some $\eta_{\mathsf{ddim}} \geq 0$. When $\eta_{\mathsf{ddim}} = 1$ and $K = T$, DDIM reduces to SMLD. But when $\eta_{\mathsf{ddim}} = 0$, the DDIM reverse process (26)

is deterministic and can be considered as a discretization of the probability-flow ODE Song et al. (2021a), which can outperform SMLD when the number of discretization steps $K$ is small (Chen et al., 2023).

For a specified number $K$ of DDIM steps (which we treat as a tuning parameter), we set the DDIM variances $\{\sigma_k^2\}_{k=1}^K$ as the geometric sequence

$$\sigma_k^2 = \sigma_{\mathsf{min}}^2 \Big(\frac{\sigma_{\mathsf{max}}^2}{\sigma_{\mathsf{min}}^2}\Big)^{\frac{k-1}{K-1}}, \quad k = 1, \dots, K \tag{28}$$

for some $\sigma_{\mathsf{min}}^2$ and $\sigma_{\mathsf{max}}^2$ that are typically chosen to match the minimum and maximum variances used to train the denoiser $\boldsymbol{d_\theta}(\cdot, \cdot)$ or score approximation $\boldsymbol{s_\theta}(\cdot, \cdot)$. So for example, if $\boldsymbol{s_\theta}(\cdot, \cdot)$ was trained over the DDPM steps $t \in \{1, \dots, T\}$ for $T = 1000$, then we would set $\sigma_{\mathsf{min}}^2 = (1 - \overline{\alpha}_1)/\overline{\alpha}_1$ and $\sigma_{\mathsf{max}}^2 = (1 - \overline{\alpha}_{1000})/\overline{\alpha}_{1000}$ with $\overline{\alpha}_t$; see (37) for additional details.

Next we discuss how we set the FIRE iteration schedule $\{N_k\}_{k=1}^K$ and variance-decrease-factor $\rho > 1$. In doing so, we have two main goals:

G1) Ensure that, at every DDIM step $k$, the denoiser's output-error variance is at most $\nu_{\mathsf{thresh}}$ at the final FIRE iteration, where $\nu_{\mathsf{thresh}}$ is some value to be determined.

G2) Meet a fixed budget of $N_{\mathsf{tot}} \triangleq \sum_{k=1}^K N_k$ total NFEs.

Note that, because the denoiser's output-error variance increases monotonically with its input-error variance, we can rephrase G1) as

G1*) Ensure that, at every DDIM step $k$, the denoiser's input-error variance is at most $\sigma_{\mathsf{thresh}}^2$ at the final FIRE iteration, where $\sigma_{\mathsf{thresh}}^2$ is some value to be determined.

Although $\sigma_{\mathsf{thresh}}^2$ could be tuned directly, it's not the most convenient option because a good search range can be difficult to construct. Instead, we tune the fraction $\delta \in [0, 1)$ of DDIM steps $k$ that use a single FIRE iteration (i.e., that use $N_k = 1$) and we set $\sigma_{\mathsf{thresh}}^2$ at the DDIM variance $\sigma_k^2$ of the first reverse-process step $k$ that uses a single FIRE iteration, i.e., $1 + \lfloor (K-1)\delta \rfloor \triangleq k_{\mathsf{thresh}}$. (Note that $k_{\mathsf{thresh}} = 1$ when $\delta = 0$ and $k_{\mathsf{thresh}} = K - 1$ for $\delta \approx 1$.) All subsequent[1] DDIM steps $k < k_{\mathsf{thresh}}$ will then automatically satisfy G1*) because $\sigma_k^2$ decreases with $k$.

To ensure that the earlier DDIM steps $k > k_{\mathsf{thresh}}$ also satisfy G1*), we need that $\sigma_k^2/\rho^{N_k - 1} \le \sigma_{\mathsf{thresh}}^2$, since $\sigma_k^2$ is the denoiser input-error variance at the first FIRE iteration and $\sigma_k^2/\rho^{N_k - 1}$ is the denoiser input-error variance at the last FIRE iteration. For a fixed $\rho > 1$, we can rewrite this inequality as

$$N_k \ge \frac{\ln \sigma_k^2 - \ln \sigma_{\mathsf{thresh}}^2}{\ln \rho} + 1 \triangleq \underline{N}_k. \tag{29}$$

Because $N_k$ is a positive integer, it suffices to choose

$$N_k = \lceil \max\{1, \underline{N}_k\} \rceil \; \forall k. \tag{30}$$

Finally, $\rho$ is chosen as the smallest value that meets the NFE budget G2) under (30). We find this value using bisection search. For a given $k_{\mathsf{thresh}}$, a lower bound on the total NFEs is $k_{\mathsf{thresh}} \cdot 1 + (K - k_{\mathsf{thresh}}) \cdot 2$. The definition of $k_{\mathsf{thresh}}$ then implies that $N_{\mathsf{tot}} \ge K(2 - \delta) + \delta - 1$ and thus $K \le (N_{\mathsf{tot}} + 1 - \delta)/(2 - \delta) \triangleq K_{\mathsf{min}}$.

In summary, for a budget of $N_{\mathsf{tot}}$ total NFEs, we treat the number of DDIM steps $K \in \{1, \dots, K_{\mathsf{min}}\}$ and the fraction of single-FIRE-iteration steps $\delta \in [0, 1)$ as tuning parameters and, from them, compute $\{\sigma_k^2\}_{k=1}^K$, $\{N_k\}_{k=1}^K$, and $\rho$. Figure 3 shows an example. The pair $(K, \delta)$ can be tuned using cross-validation. Algorithm 2 details DDIM with the FIRE approximation of $\mathrm{E}\{\boldsymbol{x}_0 | \boldsymbol{x}_k, \boldsymbol{y}\}$, which we refer to as "DDfire."

## 3.4 Relation to other methods

To solve inverse problems, a number of algorithms have been proposed that iterate denoising (possibly score-based), data-consistency (hard or soft), and renoising. Such approaches are referred to as either plug-and-play, Langevin, or diffusion methods. (Recall the discussion in Sections 1-2.) While existing approaches use white renoising, the proposed DDfire uses colored renoising that whitens the denoiser input error.

---

[1] Recall that the reverse process counts backwards, i.e., $k = K, K-1, \dots, 2, 1$.

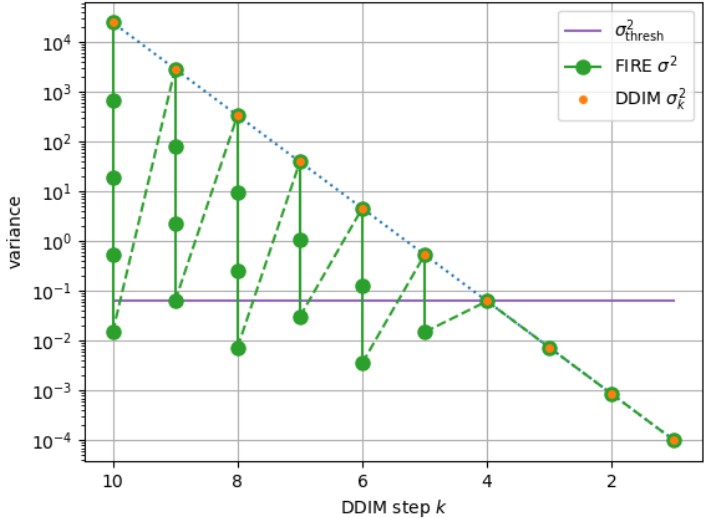

Figure 3: For an FFHQ denoiser: the geometric DDIM variances $\{\sigma_k^2\}_{k=1}^K$ versus DDIM step $k$ for $K = 10$, the $\sigma_{\text{thresh}}^2$ corresponding to a $\delta = 0.4$ fraction of single-FIRE-iteration DDIM steps, and the denoiser input variance $\sigma^2$ at each FIRE iteration of each DDIM step, for $N_{\text{tot}} = 25$ total NFEs.

---

**Algorithm 2** DDfire

**Require:** $\boldsymbol{y}, \boldsymbol{A}, \sigma_{\mathsf{y}}$ or $p_{\mathsf{y}|\mathsf{z}}, \rho, \{\sigma_k\}_{k=1}^K, \{N_k\}_{k=1}^K, \eta_{\mathsf{ddim}} \geq 0$

1: $\boldsymbol{x}_K \sim \mathcal{N}(\boldsymbol{0}, \sigma_K^2 \boldsymbol{I})$
2: **for** $k = K, K-1, \ldots, 1$ **do**
3: $\quad \widehat{\boldsymbol{x}}_{0|k} = \mathsf{FIRE}(\boldsymbol{y}, \boldsymbol{A}, *, \boldsymbol{x}_k, \sigma_k, N_k, \rho)$           ▷ *FIRE via Alg. 1 or Alg. 3*
4: $\quad \varsigma_k = \eta_{\mathsf{ddim}} \sqrt{\dfrac{\sigma_{k-1}^2(\sigma_k^2 - \sigma_{k-1}^2)}{\sigma_k^2}}$
5: $\quad \boldsymbol{x}_{k-1} = \sqrt{\dfrac{\sigma_{k-1}^2 - \varsigma_k^2}{\sigma_k^2}} \boldsymbol{x}_k + \left(1 - \sqrt{\dfrac{\sigma_{k-1}^2 - \varsigma_k^2}{\sigma_k^2}}\right)\widehat{\boldsymbol{x}}_{0|k} + \varsigma_k \boldsymbol{n}_k, \quad \boldsymbol{n}_k \sim \mathcal{N}(\boldsymbol{0}, \boldsymbol{I})$     ▷ *DDIM update*
6: **return** $\widehat{\boldsymbol{x}}_{0|k}$

---

When estimating $\boldsymbol{x}_0$ from the measurements $\boldsymbol{y}$ of (6), methods such as DDS, DiffPIR, DAPS, and SNORE use a Gaussian prior approximation of the form $\boldsymbol{x}_0 \sim \mathcal{N}(\overline{\boldsymbol{x}}, \nu \boldsymbol{I})$. But while they control $\nu$ with tuning parameters, DDfire explicitly estimates $\nu$ via (10). See App. G for a detailed comparison of DDfire to DDS, DiffPIR, and SNORE.

Although we are unaware of prior work combining EP with diffusion, there is a paper by Meng & Kabashima (2024) that applies EP to the annealed Langevin dynamics approach from Song & Ermon (2020), although for the specific task of dequantization. Like with DPS and ΠGDM, Bayes rule is used to write $\nabla_{\boldsymbol{x}} \ln p_t(\boldsymbol{x}_t|\boldsymbol{y}) = \nabla_{\boldsymbol{x}} \ln p_t(\boldsymbol{x}_t) + \nabla_{\boldsymbol{x}} \ln p_t(\boldsymbol{y}|\boldsymbol{x}_t)$ and the first term is approximated by $\boldsymbol{s}_{\boldsymbol{\theta}}(\boldsymbol{x}_t, t)$ (recall Sec. 2). The second term is approximated with multiple EP iterations, each of which solves a linear system using an SVD. DDfire differs in using colored renoising, an SVD-free option, and DDIM diffusion, which allows far fewer NFEs.

## 4 Numerical experiments

We use $256 \times 256$ FFHQ (Karras et al., 2019) and ImageNet (Deng et al., 2009) datasets with pretrained diffusion models from Chung et al. (2023a) and Dhariwal & Nichol (2021), respectively. As linear inverse problems, we consider box inpainting with a $128 \times 128$ mask, Gaussian deblurring using a $61 \times 61$ blur kernel with 3-pixel standard deviation, motion deblurring using a $61 \times 61$ blur kernel generated using Borodenko (2020) with intensity 0.5, and $4\times$ bicubic super-resolution. We compare to DDRM (Kawar et al., 2022a),

Table 1: DDfire ablation results for noisy FFHQ box inpainting with $\sigma_{\mathsf{y}} = 0.05$ at 1000 NFEs.

| Method | PSNR↑ | LPIPS↓ | Runtime |
|---|---|---|---|
| DDfire | 24.31 | 0.1127 | 34.37s |
| DDfire w/o renoising | 18.48 | 0.2349 | 34.37s |
| DDfire w/o colored renoising | 23.64 | 0.1553 | 34.37s |
| DDfire w/ stochastic denoising | 24.30 | 0.1143 | 34.37s |
| DDfire w/o estimating $\nu$ | 23.02 | 0.1755 | 34.37s |
| DDfire w/o CG early stopping | 24.31 | 0.1127 | 52.12s |
| DDfire w/ SVD | 24.31 | 0.1124 | 30.97s |

DiffPIR (Zhu et al., 2023), ΠGDM (Song et al., 2023), DDS Chung et al. (2024a), DPS (Chung et al., 2023a), RED-diff (Mardani et al., 2024), and DAPS (Zhang et al., 2025).

We also consider phase retrieval with the shot-noise corruption mechanism from Metzler et al. (2018) for both oversampled Fourier (OSF) and coded diffraction pattern (CDP) (Candès et al., 2015) $\boldsymbol{A}$ at 4× oversampling with $\alpha_{\mathsf{shot}} = 8$ and 45, respectively. Here, $\alpha_{\mathsf{shot}}$ is the shot-noise strength, as detailed in App. H. We compare to prDeep (Metzler et al., 2018), DOLPH (Shoushtari et al., 2023), DPS, RED-diff, DAPS, and the classical hybrid input-output (HIO) algorithm (Fienup, 1982), using $p_{\mathsf{y}|\mathsf{z}}(y|z) = \mathcal{N}(y; |z|, \sigma_{\mathsf{y}}^2)$ for all algorithms that accept a likelihood function.

Unless specified otherwise, DDfire was configured as follows. For the linear inverse problems, CG is used (no SVD), 1000 NFEs are used without stochastic denoising, and the $(K, \delta)$ hyperparameters are tuned to minimize LPIPS (Zhang et al., 2018) on a 100-sample validation set (see Table 5). For phase retrieval, we set NFEs to 800 for OSF to fairly compare to prDeep, while only 100 NFEs were used for CDP as further steps offered no improvement. Stochastic damping was applied in both cases, and $(K, \delta)$ were the hand-tuned values in Table 6. Neither CG nor SVD are needed since (11) can be solved in analytically. Appendix H contains additional details on the implementation of DDfire and the competing methods.

### 4.1 Ablation study

We first perform an ablation study on the SLM-DDfire design choices in Sec. 3 using noisy FFHQ box inpainting and a 100-image validation set. The results are summarized in Table 1. We first see that both PSNR and LPIPS suffer significantly when FIRE is run without renoising (i.e., $\boldsymbol{c} = \boldsymbol{0}$ in line 13 in Alg. 1). Similarly, renoising using white noise (i.e., $\boldsymbol{c} \sim \mathcal{N}(\boldsymbol{0}, \sigma^2\boldsymbol{I})$ in line 13 of Alg. 1) gives noticeably worse PSNR and LPIPS than the proposed colored noise. Using stochastic denoising gives nearly identical performance to plain denoising (i.e., $\widehat{\nu}_\phi(\sigma) = 0$ in line 3 of Alg. 1), and so we use plain denoising by default with linear inverse problems. A more significant degradation results when the denoiser output-error variance $\nu$ is not adapted to $\overline{\boldsymbol{x}}$ in line 4 of Alg. 1 but set at the data-average value $\widehat{\nu}_\phi(\sigma)$. On the other hand, when CG doesn't use early stopping (as described in App. D) the runtime increases without improving PSNR or LPIPS. Thus, we use early stopping by default. Finally, using an SVD instead of CG, which also avoids the noise approximation in (17), gives essentially identical PSNR and LPIPS but with a slightly faster runtime. Figure 4 shows another LPIPS/runtime comparison of the SVD and CG versions of DDfire.

### 4.2 PSNR, LPIPS, and FID results

For noisy linear inverse problems, Tables 2–3 show PSNR, LPIPS, and FID (Heusel et al., 2017) on a 1000-sample test set for FFHQ and ImageNet data, respectively. DDRM was not applied to motion deblurring due to the lack of an SVD. Tables 2–3 show that DDfire wins in most cases and otherwise performs well.

Fig. 5 shows image examples for inpainting, motion deblurring, Gaussian deblurring, and 4× super-resolution on ImageNet. The zoomed regions show that DDfire did a better job recovering fine details. Additional examples can be found in Fig. 15.

For OSF and CDP phase retrieval, Table 4 shows PSNR, LPIPS, and FID on a 1000-sample test set for FFHQ and ImageNet data. The table shows that DDfire outperformed the competitors in all cases. With

Table 2: Noisy FFHQ results with measurement noise standard deviation $\sigma_y = 0.05$.

| Model | Inpaint (box) | | | Deblur (Gaussian) | | | Deblur (Motion) | | | 4× Super-resolution | | |
|---|---|---|---|---|---|---|---|---|---|---|---|---|
| | PSNR↑ | LPIPS↓ | FID↓ | PSNR↑ | LPIPS↓ | FID↓ | PSNR↑ | LPIPS↓ | FID↓ | PSNR↑ | LPIPS↓ | FID↓ |
| DDRM | 21.71 | 0.1551 | 40.61 | 25.35 | 0.2223 | 51.70 | - | - | - | **27.32** | 0.1864 | 45.82 |
| DiffPIR | 22.43 | 0.1883 | 31.98 | 24.56 | 0.2394 | 34.82 | 26.91 | 0.1952 | 26.67 | 24.89 | 0.2486 | 32.33 |
| ΠGDM | 21.41 | 0.2009 | 44.41 | 23.66 | 0.2525 | 45.34 | 25.14 | 0.2082 | 41.95 | 24.40 | 0.2520 | 51.41 |
| DDS | 20.28 | 0.1481 | 30.23 | 26.74 | 0.1648 | 25.47 | 27.52 | 0.1503 | 27.59 | 26.71 | 0.1852 | 27.09 |
| DPS | 22.54 | 0.1368 | 35.69 | 25.70 | 0.1774 | 25.18 | 26.74 | 0.1655 | 27.17 | 26.30 | 0.1850 | 27.38 |
| RED-diff | 23.58 | 0.1883 | 48.86 | 26.99 | 0.2081 | 38.82 | 16.47 | 0.5074 | 128.68 | 25.61 | 0.3569 | 70.86 |
| DAPS | 23.61 | 0.1415 | 31.51 | 26.97 | 0.1827 | 31.10 | 27.13 | 0.1718 | 30.74 | 26.91 | 0.1885 | 30.83 |
| DDfire | **24.75** | **0.1101** | **25.26** | **27.10** | **0.1533** | **24.97** | **28.14** | **0.1374** | **26.12** | 27.13 | **0.1650** | **25.73** |

Table 3: Noisy ImageNet results with measurement noise standard deviation $\sigma_y = 0.05$.

| Model | Inpaint (box) | | | Deblur (Gaussian) | | | Deblur (Motion) | | | 4× Super-resolution | | |
|---|---|---|---|---|---|---|---|---|---|---|---|---|
| | PSNR↑ | LPIPS↓ | FID↓ | PSNR↑ | LPIPS↓ | FID↓ | PSNR↑ | LPIPS↓ | FID↓ | PSNR↑ | LPIPS↓ | FID↓ |
| DDRM | 18.24 | 0.2423 | 67.47 | 22.56 | 0.3454 | 68.78 | - | - | - | **24.49** | 0.2777 | 64.68 |
| DiffPIR | 18.03 | 0.2860 | 65.55 | 21.31 | 0.3683 | 56.35 | 24.36 | 0.2888 | 54.11 | 23.31 | 0.3383 | 63.48 |
| ΠGDM | 17.69 | 0.3303 | 86.36 | 20.87 | 0.4191 | 75.43 | 22.15 | 0.3591 | 70.91 | 21.25 | 0.4149 | 78.57 |
| DDS | 16.68 | 0.2222 | 63.07 | 23.14 | 0.2684 | 50.84 | 23.34 | 0.2674 | 50.08 | 23.03 | 0.3011 | 52.13 |
| DPS | 18.23 | 0.2314 | 59.10 | 21.30 | 0.3393 | 50.46 | 21.77 | 0.3307 | 80.27 | 23.38 | 0.2904 | 49.86 |
| RED-diff | 18.95 | 0.2909 | 108.88 | 23.45 | 0.3190 | 65.65 | 15.21 | 0.5647 | 198.74 | 22.99 | 0.3858 | 83.06 |
| DAPS | 19.99 | 0.2199 | 61.53 | **23.91** | 0.2863 | 56.87 | 24.58 | 0.2722 | 54.83 | 24.04 | 0.2729 | 55.54 |
| DDfire | **20.39** | **0.1915** | **55.54** | 23.71 | **0.2353** | **50.05** | **24.59** | **0.2314** | **49.25** | 23.58 | **0.2629** | **49.67** |

Table 4: Noisy phase retrieval results

| Model | FFHQ OSF | | | FFHQ CDP | | | ImageNet OSF | | | ImageNet CDP | | |
|---|---|---|---|---|---|---|---|---|---|---|---|---|
| | PSNR↑ | LPIPS↓ | FID↓ | PSNR↑ | LPIPS↓ | FID↓ | PSNR↑ | LPIPS↓ | FID↓ | PSNR↑ | LPIPS↓ | FID↓ |
| HIO | 23.66 | 0.4706 | 130.6 | 17.59 | 0.5430 | 84.87 | 21.53 | 0.4584 | 111.5 | 17.65 | 0.4527 | 69.01 |
| DOLPH | 14.73 | 0.7220 | 389.9 | 25.76 | 0.1686 | 32.93 | 14.33 | 0.6844 | 258.6 | 26.15 | 0.2256 | 48.26 |
| DPS | 23.63 | 0.2908 | 53.91 | 29.19 | 0.1394 | 27.87 | 16.69 | 0.5314 | 140.2 | 27.21 | 0.1799 | 45.98 |
| RED-diff | 25.47 | 0.2828 | 65.74 | 28.75 | 0.1734 | 28.87 | 18.41 | 0.4385 | 108.0 | 26.87 | 0.2173 | 48.12 |
| DAPS | 24.10 | 0.2891 | 57.73 | 28.26 | 0.1927 | 34.97 | 17.62 | 0.4934 | 121.8 | 21.76 | 0.3332 | 54.17 |
| prDeep | 30.90 | 0.1132 | 31.51 | 19.24 | 0.4183 | 59.44 | 26.16 | 0.1977 | 57.86 | 19.51 | 0.3934 | 59.34 |
| DDfire | **33.56** | **0.0691** | **28.94** | **30.16** | **0.1186** | **23.30** | **29.93** | **0.1640** | **56.99** | **28.91** | **0.1377** | **44.35** |

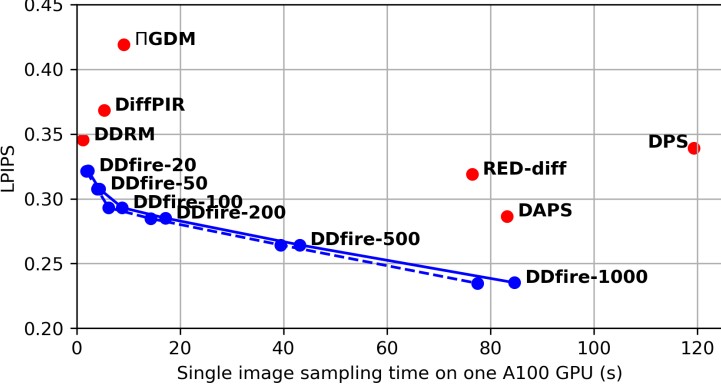

Figure 4: LPIPS vs. single image sampling time for noisy Gaussian deblurring on an A100 GPU. The evaluation used 1000 ImageNet images. Solid line: DDfire with CG for various numbers of NFEs. Dashed line: DDfire with SVD.

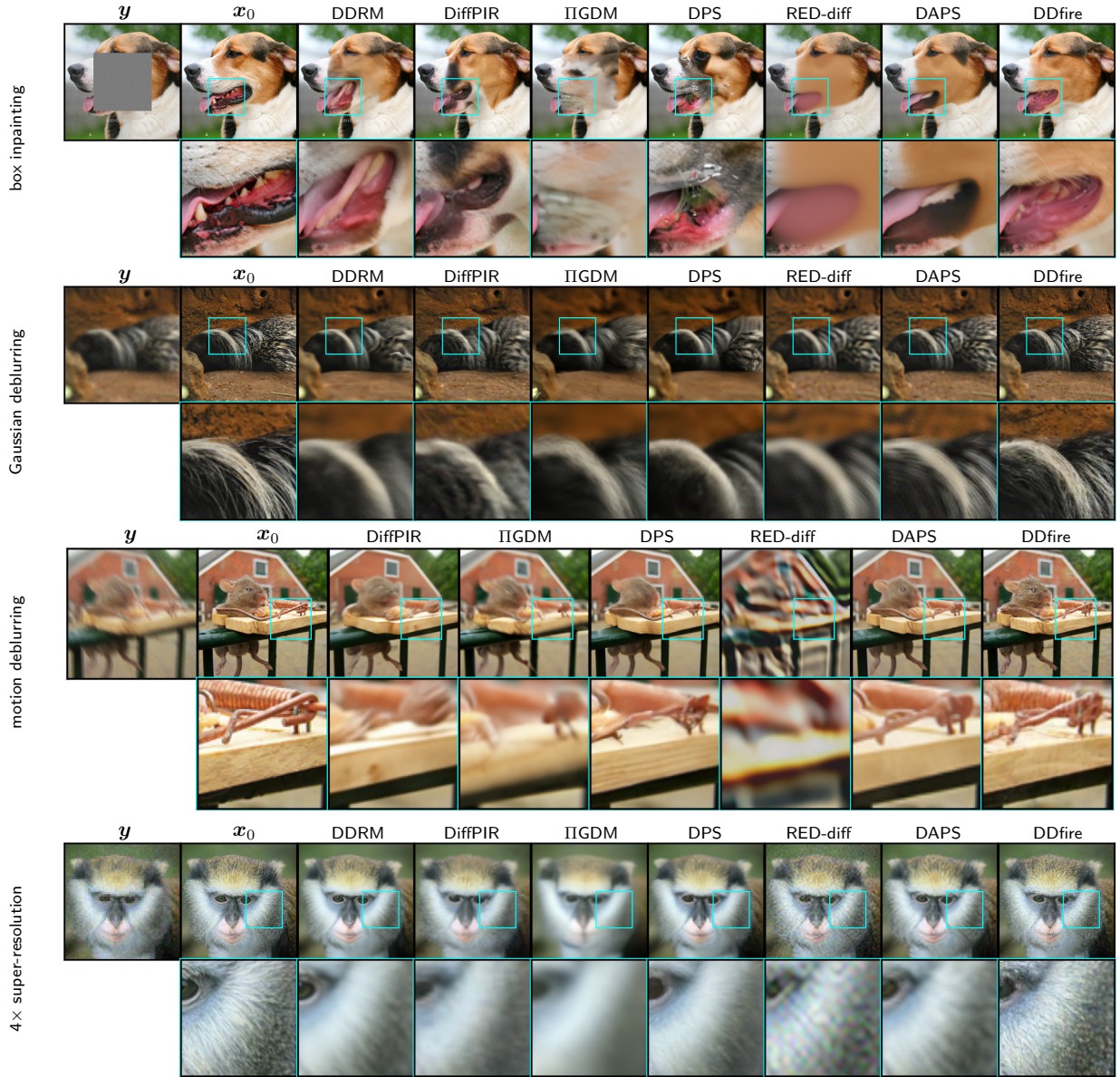

Figure 5: Example recoveries from noisy linear inverse problems with ImageNet images.

FFHQ OSF, for example, DDfire outperformed the best competitor (prDeep) by 2.6 dB and the second best (RED-diff) by 8.1 dB in PSNR. Example reconstructions can be found in Fig. 14.

## 4.3 Runtime results

Figure 4 shows LPIPS vs. average runtime (in seconds on an A100 GPU) to generate a single image for noisy Gaussian deblurring on the 1000-sample ImageNet test set. The figure shows that DDfire gives a better performance/complexity tradeoff than the competitors. It also shows that DDfire is approximately 1.5 times faster than DPS when both are run at 1000 NFEs, due to DPS's use of backpropagation.

Figure 11 shows LPIPS vs. runtime for noisy OSF phase retrieval on the 1000-sample FFHQ test set, showing that DDfire gives a significantly better performance/complexity tradeoff than all diffusion-based competitors.

## 5 Conclusion

To solve linear inverse problems, we proposed the Fast Iterative Renoising (FIRE) algorithm, which can be interpreted as the HQS plug-and-play algorithm with a colored renoising step that aims to whiten the denoiser input error. We then extended the linear FIRE algorithm to the generalized-linear case using expectation propagation (EP). Since these FIRE algorithms approximate the measurement-conditional denoiser $\mathrm{E}\{\boldsymbol{x}_0|\boldsymbol{x}_t,\boldsymbol{y}\}$, or equivalently the measurement-conditional score $\nabla_{\boldsymbol{x}} \ln p_t(\boldsymbol{x}_t|\boldsymbol{y})$, they can be readily combined with DDIM for diffusion posterior sampling, giving the "DDfire" algorithm. Experiments on box inpainting, Gaussian and motion deblurring, and $4\times$ super-resolution with FFHQ and ImageNet images show DDfire outperforming DDRM, ΠGDM, DDS, DiffPIR, DPS, RED-diff, and DAPS in PSNR, LPIPS, and FID metrics in nearly all cases. Experiments on noisy FFHQ and ImageNet phase retrieval (both OSF and CDP versions) show DDfire outperforming HIO, prDeep, DOLPH, DPS, RED-diff, and DAPS in all cases. Finally, DDfire offers fast inference, with better LPIPS-versus-runtime curves than the competitors.

## Acknowledgements

This work was funded in part by the National Institutes of Health under grant R01-EB029957.

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

## A    VP formulation

In the main text, we describe DDfire for the VE SDE formulation from Song et al. (2021b) and the corresponding SMLD discretization from Song & Ermon (2019). Here, we describe it for the VP SDE from Song et al. (2021b) and the corresponding DDPM discretization from Ho et al. (2020).

From Song et al. (2021b), the general SDE forward process can be written as

$$\mathrm{d}\boldsymbol{x} = \boldsymbol{f}(\boldsymbol{x}, t)\,\mathrm{d}t + g(t)\,\mathrm{d}\boldsymbol{w} \tag{31}$$

for some choices of $\boldsymbol{f}(\cdot, \cdot)$ and $g(\cdot)$, where $\mathrm{d}\boldsymbol{w}$ is the standard Wiener process (i.e., Brownian motion). The reverse process can then be described by

$$\mathrm{d}\boldsymbol{x} = \big(\boldsymbol{f}(\boldsymbol{x}, t) - g^2(t)\nabla_{\boldsymbol{x}}\ln p_t(\boldsymbol{x})\big)\,\mathrm{d}t + g(t)\,\mathrm{d}\overline{\boldsymbol{w}}, \tag{32}$$

where $p_t(\cdot)$ is the distribution of $\boldsymbol{x}$ at time $t$ and $\mathrm{d}\overline{\boldsymbol{w}}$ is the reverse Wiener process. In the VE-SDE, $\boldsymbol{f}(\boldsymbol{x}, t) = \boldsymbol{0}$ and $g(t) = \sqrt{\mathrm{d}[\sigma^2(t)]/\,\mathrm{d}t}$ for some variance schedule $\sigma^2(t)$, but in the VP-SDE, $\boldsymbol{f}(\boldsymbol{x}, t) = -\frac{1}{2}\beta(t)\boldsymbol{x}$ and $g(t) = \sqrt{\beta(t)}$ for some variance schedule $\beta(t)$. When discretized to $t \in \{0, 1, \ldots, T\}$, the VP forward process becomes

$$\widetilde{\boldsymbol{x}}_t = \sqrt{1-\beta_t}\widetilde{\boldsymbol{x}}_{t-1} + \sqrt{\beta_t}\widetilde{\boldsymbol{w}}_{t-1} \tag{33}$$

with i.i.d. $\{\widetilde{\boldsymbol{w}}_t\} \sim \mathcal{N}(\boldsymbol{0}, \boldsymbol{I})$, so that

$$\widetilde{\boldsymbol{x}}_t = \sqrt{\overline{\alpha}_t}\boldsymbol{x}_0 + \sqrt{1-\overline{\alpha}_t}\widetilde{\boldsymbol{\epsilon}}_t \tag{34}$$

with $\alpha_t \triangleq 1-\beta_t$, $\overline{\alpha}_t = \prod_{s=1}^{t}\alpha_s$, and $\widetilde{\boldsymbol{\epsilon}}_t \sim \mathcal{N}(\boldsymbol{0}, \boldsymbol{I})$. Throughout, we write the VP quantities with tildes to distinguish them from the VE quantities. The DDPM reverse process then takes the form

$$\widetilde{\boldsymbol{x}}_{t-1} = \frac{1}{\sqrt{\alpha_t}}\big(\widetilde{\boldsymbol{x}}_t + \beta_t\nabla_{\boldsymbol{x}_t}\ln p(\widetilde{\boldsymbol{x}}_t)\big) + \Sigma_t\widetilde{\boldsymbol{n}}_t \quad \text{with} \quad \Sigma_t^2 \triangleq \frac{1-\overline{\alpha}_{t-1}}{1-\overline{\alpha}_t}\beta_t \tag{35}$$

and is typically initialized at $\widetilde{\boldsymbol{x}}_T \sim \mathcal{N}(\boldsymbol{0}, \boldsymbol{I})$. By rewriting (34) as

$$\frac{1}{\sqrt{\overline{\alpha}_t}}\widetilde{\boldsymbol{x}}_t = \boldsymbol{x}_0 + \sqrt{\frac{1-\overline{\alpha}_t}{\overline{\alpha}_t}}\widetilde{\boldsymbol{\epsilon}}_t \tag{36}$$

and comparing it to (3), we recognize the VP/VE relationships

$$\frac{1}{\sqrt{\overline{\alpha}_t}}\widetilde{\boldsymbol{x}}_t = \boldsymbol{x}_t \quad \text{and} \quad \frac{1-\overline{\alpha}_t}{\overline{\alpha}_t} = \sigma_t^2 \ \Leftrightarrow\ \overline{\alpha}_t = \frac{1}{1+\sigma_t^2}. \tag{37}$$

Furthermore, assuming that $\overline{\alpha}_T \ll 1$, the VP initialization $\widetilde{\boldsymbol{x}}_T \sim \mathcal{N}(\boldsymbol{0}, \boldsymbol{I})$ is well approximated by the VE initialization $\boldsymbol{x}_T \sim \mathcal{N}(\boldsymbol{0}, \sigma_T^2\boldsymbol{I})$.

## B    DDIM details for VP

The DDIM reverse process from Song et al. (2021a) provides an alternative to the DDPM reverse process that offers a flexible level of stochasticity. When describing VP DDIM, we will write the quantities as $\widetilde{\boldsymbol{x}}_k, \widetilde{\boldsymbol{\epsilon}}_k, \widetilde{\boldsymbol{n}}_k, \overline{\alpha}_k$ to distinguish them from the corresponding VP DDPM quantities $\widetilde{\boldsymbol{x}}_t, \widetilde{\boldsymbol{\epsilon}}_t, \widetilde{\boldsymbol{n}}_t, \overline{\alpha}_t$, and we will write the total number of steps as $K$. Like (34), DDIM is built around the model

$$\widetilde{\boldsymbol{x}}_k = \sqrt{\overline{\alpha}_k}\boldsymbol{x}_0 + \sqrt{1-\overline{\alpha}_k}\widetilde{\boldsymbol{\epsilon}}_k, \quad \widetilde{\boldsymbol{\epsilon}}_k \sim \mathcal{N}(\boldsymbol{0}, \boldsymbol{I}). \tag{38}$$

Adapting the first two equations from Song et al. (2021a, App.D.3) to our notation, we have

$$\widetilde{\boldsymbol{x}}_{k-1} = \sqrt{\overline{\alpha}_{k-1}}\bigg(\frac{\widetilde{\boldsymbol{x}}_k - \sqrt{1-\overline{\alpha}_k}\,\mathrm{E}\{\widetilde{\boldsymbol{\epsilon}}_k|\widetilde{\boldsymbol{x}}_k, \boldsymbol{y}\}}{\sqrt{\overline{\alpha}_k}}\bigg) + \sqrt{1-\overline{\alpha}_{k-1} - \widetilde{\varsigma}_k^2}\,\mathrm{E}\{\widetilde{\boldsymbol{\epsilon}}_k|\widetilde{\boldsymbol{x}}_k, \boldsymbol{y}\} + \widetilde{\varsigma}_k\widetilde{\boldsymbol{n}}_k \tag{39}$$

$$\widetilde{\varsigma}_k \triangleq \eta_{\mathsf{ddim}}\sqrt{\frac{1-\overline{\alpha}_{k-1}}{1-\overline{\alpha}_k}}\sqrt{1 - \frac{\overline{\alpha}_k}{\overline{\alpha}_{k-1}}} \tag{40}$$

with $\widetilde{\boldsymbol{n}}_k \sim \mathcal{N}(\mathbf{0}, \boldsymbol{I})$ and tunable $\eta_{\mathsf{ddim}} \geq 0$. When $\eta_{\mathsf{ddim}} = 1$ and $K = T$, DDIM reduces to DDPM. But when $\eta_{\mathsf{ddim}} = 0$, the reverse process is deterministic. In fact, it can be considered a discretization of the probability flow ODE (Song et al., 2021a), which often works much better than the SDE when the number of discretization steps $K$ is small. We now write (39) in a simpler form. Applying $\mathrm{E}\{\cdot|\widetilde{\boldsymbol{x}}_k, \boldsymbol{y}\}$ to both sides of (38) gives

$$\widetilde{\boldsymbol{x}}_k = \sqrt{\overline{\alpha}_k}\,\mathrm{E}\{\boldsymbol{x}_0|\widetilde{\boldsymbol{x}}_k, \boldsymbol{y}\} + \sqrt{1-\overline{\alpha}_k}\,\mathrm{E}\{\widetilde{\boldsymbol{\epsilon}}_k|\widetilde{\boldsymbol{x}}_k, \boldsymbol{y}\} \quad \Leftrightarrow \quad \mathrm{E}\{\widetilde{\boldsymbol{\epsilon}}_k|\widetilde{\boldsymbol{x}}_k, \boldsymbol{y}\} = \frac{\widetilde{\boldsymbol{x}}_k - \sqrt{\overline{\alpha}_k}\,\mathrm{E}\{\boldsymbol{x}_0|\widetilde{\boldsymbol{x}}_k, \boldsymbol{y}\}}{\sqrt{1-\overline{\alpha}_k}}, \tag{41}$$

and plugging (41) into (39) gives

$$\widetilde{\boldsymbol{x}}_{k-1} = \sqrt{\overline{\alpha}_{k-1}}\,\mathrm{E}\{\boldsymbol{x}_0|\widetilde{\boldsymbol{x}}_k, \boldsymbol{y}\} + \widetilde{\varsigma}_k\widetilde{\boldsymbol{n}}_k + \sqrt{1-\overline{\alpha}_{k-1}-\widetilde{\varsigma}_k^2}\left(\frac{\widetilde{\boldsymbol{x}}_k - \sqrt{\overline{\alpha}_k}\,\mathrm{E}\{\boldsymbol{x}_0|\widetilde{\boldsymbol{x}}_k, \boldsymbol{y}\}}{\sqrt{1-\overline{\alpha}_k}}\right) \tag{42}$$

$$= \sqrt{\overline{\alpha}_{k-1}}\,\mathrm{E}\{\boldsymbol{x}_0|\widetilde{\boldsymbol{x}}_k, \boldsymbol{y}\} + \widetilde{\varsigma}_k\widetilde{\boldsymbol{n}}_k + \widetilde{h}_k\big(\widetilde{\boldsymbol{x}}_k - \sqrt{\overline{\alpha}_k}\,\mathrm{E}\{\boldsymbol{x}_0|\widetilde{\boldsymbol{x}}_k, \boldsymbol{y}\}\big) \tag{43}$$

$$= \widetilde{h}_k\widetilde{\boldsymbol{x}}_k + \widetilde{g}_k\,\mathrm{E}\{\boldsymbol{x}_0|\widetilde{\boldsymbol{x}}_k, \boldsymbol{y}\} + \widetilde{\varsigma}_k\widetilde{\boldsymbol{n}}_k \tag{44}$$

for

$$\widetilde{h}_k \triangleq \sqrt{\frac{1-\overline{\alpha}_{k-1}-\widetilde{\varsigma}_k^2}{1-\overline{\alpha}_k}} \quad \text{and} \quad \widetilde{g}_k \triangleq \sqrt{\overline{\alpha}_{k-1}} - \widetilde{h}_k\sqrt{\overline{\alpha}_k}. \tag{45}$$

Thus the VP DDIM reverse process can be described by (40), (44), and (45) with $\widetilde{\boldsymbol{n}}_k \sim \mathcal{N}(\mathbf{0}, \boldsymbol{I})\ \forall k$ and initialization $\widetilde{\boldsymbol{x}}_K \sim \mathcal{N}(\mathbf{0}, \boldsymbol{I})$.

## C  DDIM details for VE

We now provide the details for the VE version of DDIM. Starting with the VP DDIM reverse process (44), we can divide both sides by $\sqrt{\overline{\alpha}_{k-1}}$ to get

$$\frac{\widetilde{\boldsymbol{x}}_{k-1}}{\sqrt{\overline{\alpha}_{k-1}}} = \frac{\widetilde{h}_k\sqrt{\overline{\alpha}_k}}{\sqrt{\overline{\alpha}_{k-1}}}\frac{\widetilde{\boldsymbol{x}}_k}{\sqrt{\overline{\alpha}_k}} + \frac{\widetilde{g}_k}{\sqrt{\overline{\alpha}_{k-1}}}\,\mathrm{E}\{\boldsymbol{x}_0|\widetilde{\boldsymbol{x}}_k, \boldsymbol{y}\} + \frac{\widetilde{\varsigma}_k}{\sqrt{\overline{\alpha}_{k-1}}}\widetilde{\boldsymbol{n}}_k \tag{46}$$

and leveraging the VP-to-VE relationship (37) to write

$$\boldsymbol{x}_{k-1} = h_k\boldsymbol{x}_k + g_k\,\mathrm{E}\{\boldsymbol{x}_0|\boldsymbol{x}_k, \boldsymbol{y}\} + \varsigma_k\boldsymbol{n}_k \quad \text{with} \quad h_k = \frac{\widetilde{h}_k\sqrt{\overline{\alpha}_k}}{\sqrt{\overline{\alpha}_{k-1}}}, \quad g_k = \frac{\widetilde{g}_k}{\sqrt{\overline{\alpha}_{k-1}}}, \quad \varsigma_k = \frac{\widetilde{\varsigma}_k}{\sqrt{\overline{\alpha}_{k-1}}} \tag{47}$$

with $\boldsymbol{n}_k \sim \mathcal{N}(\mathbf{0}, \boldsymbol{I})\ \forall k$ and initialization $\boldsymbol{x}_K \sim \mathcal{N}(\mathbf{0}, \sigma_K^2\boldsymbol{I})$. Plugging $\widetilde{g}_k$ from (45) into (47), we find

$$g_k = \frac{\sqrt{\overline{\alpha}_{k-1}} - \widetilde{h}_k\sqrt{\overline{\alpha}_k}}{\sqrt{\overline{\alpha}_{k-1}}} = 1 - h_k. \tag{48}$$

Then plugging (40) into (47), we find

$$\varsigma_k = \frac{\eta_{\mathsf{ddim}}}{\sqrt{\overline{\alpha}_{k-1}}}\sqrt{\frac{1-\overline{\alpha}_{k-1}}{1-\overline{\alpha}_k}}\sqrt{1-\frac{\overline{\alpha}_k}{\overline{\alpha}_{k-1}}} = \eta_{\mathsf{ddim}}\sqrt{\frac{1}{\overline{\alpha}_{k-1}}\frac{1-\overline{\alpha}_{k-1}}{1-\overline{\alpha}_k}\left(1-\frac{\overline{\alpha}_k}{\overline{\alpha}_{k-1}}\right)} \tag{49}$$

$$= \eta_{\mathsf{ddim}}\sqrt{\frac{1-\overline{\alpha}_{k-1}}{\overline{\alpha}_{k-1}}\frac{\overline{\alpha}_k}{1-\overline{\alpha}_k}\left(\frac{1}{\overline{\alpha}_k}-\frac{1}{\overline{\alpha}_{k-1}}\right)} = \eta_{\mathsf{ddim}}\sqrt{\frac{\sigma_{k-1}^2}{\sigma_k^2}\left([1+\sigma_k^2]-[1+\sigma_{k-1}^2]\right)} \tag{50}$$

$$= \eta_{\mathsf{ddim}}\sqrt{\frac{\sigma_{k-1}^2(\sigma_k^2-\sigma_{k-1}^2)}{\sigma_k^2}}. \tag{51}$$

Finally, noting from (37), (47), and (51) that

$$\frac{\widetilde{\varsigma}_k^2}{1-\overline{\alpha}_{k-1}} = \varsigma_k^2\frac{\overline{\alpha}_{k-1}}{1-\overline{\alpha}_{k-1}} = \frac{\varsigma_k^2}{\sigma_{k-1}^2} = \frac{\eta_{\mathsf{ddim}}^2}{\sigma_{k-1}^2}\frac{\sigma_{k-1}^2(\sigma_k^2-\sigma_{k-1}^2)}{\sigma_k^2} = \eta_{\mathsf{ddim}}^2\left(1-\frac{\sigma_{k-1}^2}{\sigma_k^2}\right), \tag{52}$$

we plug $\widetilde{h}_k$ from (45) into (47) to find

$$h_k = \sqrt{\frac{\overline{\alpha}_k}{\overline{\alpha}_{k-1}} \frac{1 - \overline{\alpha}_{k-1} - \widetilde{\varsigma}_k^2}{1 - \overline{\alpha}_k}} = \sqrt{\frac{\overline{\alpha}_k}{1 - \overline{\alpha}_k} \frac{1 - \overline{\alpha}_{k-1} - \widetilde{\varsigma}_k^2}{\overline{\alpha}_{k-1}}} \tag{53}$$

$$= \sqrt{\frac{\overline{\alpha}_k}{1 - \overline{\alpha}_k} \frac{1 - \overline{\alpha}_{k-1}}{\overline{\alpha}_{k-1}} \left(1 - \frac{\widetilde{\varsigma}_k^2}{1 - \overline{\alpha}_{k-1}}\right)} = \sqrt{\frac{\sigma_{k-1}^2}{\sigma_k^2} \left(1 - \eta_{\mathsf{ddim}}^2 \left(1 - \frac{\sigma_{k-1}^2}{\sigma_k^2}\right)\right)} \tag{54}$$

$$= \sqrt{\frac{\sigma_{k-1}^2}{\sigma_k^2} \left(1 - \frac{\varsigma_k^2}{\sigma_{k-1}^2}\right)} = \sqrt{\frac{\sigma_{k-1}^2}{\sigma_k^2} - \frac{\varsigma_k^2}{\sigma_k^2}} = \sqrt{\frac{\sigma_{k-1}^2 - \varsigma_k^2}{\sigma_k^2}}. \tag{55}$$

The VE DDIM reverse process is summarized in (26)-(27).

## D Speeding up CG

In this section, we describe a small modification to FIRE that can help to speed up the CG step. When CG is used to solve (11), its convergence speed is determined by the condition number of $\boldsymbol{A}^\mathsf{T}\boldsymbol{A} + (\sigma_{\mathsf{y}}^2/\nu)\boldsymbol{I}$ (Luenberger & Ye, 2016). Thus CG can converge slowly when $\sigma_{\mathsf{y}}^2/\nu$ is small, which can happen in early DDfire iterations. To speed up CG, we propose to solve (11) using $\widehat{\sigma}_{\mathsf{y}}$ in place of $\sigma_{\mathsf{y}}$, for some $\widehat{\sigma}_{\mathsf{y}} > \sigma_{\mathsf{y}}$. Since the condition number of $\boldsymbol{A}^\mathsf{T}\boldsymbol{A} + (\widehat{\sigma}_{\mathsf{y}}^2/\nu)\boldsymbol{I}$ is at most $\nu s_{\mathsf{max}}^2/\widehat{\sigma}_{\mathsf{y}}^2 + 1$, we can guarantee a conditional number of at most $10\,001$ by setting

$$\widehat{\sigma}_{\mathsf{y}}^2 = \nu s_{\mathsf{max}}^2 \max\{10^{-4}, \sigma_{\mathsf{y}}^2/(\nu s_{\mathsf{max}}^2)\}. \tag{56}$$

Although using $\widehat{\sigma}_{\mathsf{y}} > \sigma_{\mathsf{y}}$ in (11) will degrade the MSE of $\widehat{\boldsymbol{x}}$, the degradation is partially offset by the fact that less noise will be added when renoising $\boldsymbol{r}$. In any case, the modified (11) can be written as

$$\widehat{\boldsymbol{x}} = (\boldsymbol{A}^\mathsf{T}\boldsymbol{A}/\widehat{\sigma}_{\mathsf{y}}^2 + \boldsymbol{I}/\nu)^{-1}(\boldsymbol{A}^\mathsf{T}\boldsymbol{y}/\widehat{\sigma}_{\mathsf{y}}^2 + \overline{\boldsymbol{x}}/\nu) \tag{57}$$

$$= (\boldsymbol{A}^\mathsf{T}\boldsymbol{A}/\widehat{\sigma}_{\mathsf{y}}^2 + \boldsymbol{I}/\nu)^{-1}(\boldsymbol{A}^\mathsf{T}[\boldsymbol{A}\boldsymbol{x}_0 + \sigma_{\mathsf{y}}\boldsymbol{w}]/\widehat{\sigma}_{\mathsf{y}}^2 + [\boldsymbol{x}_0 - \sqrt{\nu}\boldsymbol{e}]/\nu) \tag{58}$$

$$= \boldsymbol{x}_0 + (\boldsymbol{A}^\mathsf{T}\boldsymbol{A}/\widehat{\sigma}_{\mathsf{y}}^2 + \boldsymbol{I}/\nu)^{-1}(\boldsymbol{A}^\mathsf{T}\boldsymbol{w}\sigma_{\mathsf{y}}/\widehat{\sigma}_{\mathsf{y}}^2 - \boldsymbol{e}/\sqrt{\nu}), \tag{59}$$

in which case $\widehat{\boldsymbol{x}} \sim \mathcal{N}(\boldsymbol{x}_0, \boldsymbol{C})$ with covariance

$$\boldsymbol{C} = (\boldsymbol{A}^\mathsf{T}\boldsymbol{A}/\widehat{\sigma}_{\mathsf{y}}^2 + \boldsymbol{I}/\nu)^{-1}(\boldsymbol{A}^\mathsf{T}\boldsymbol{A}\sigma_{\mathsf{y}}^2/\widehat{\sigma}_{\mathsf{y}}^4 + \boldsymbol{I}/\nu)(\boldsymbol{A}^\mathsf{T}\boldsymbol{A}/\widehat{\sigma}_{\mathsf{y}}^2 + \boldsymbol{I}/\nu)^{-1} \tag{60}$$

$$= (\boldsymbol{V}\boldsymbol{S}^\mathsf{T}\boldsymbol{S}\boldsymbol{V}^\mathsf{T}/\widehat{\sigma}_{\mathsf{y}}^2 + \boldsymbol{I}/\nu)^{-1}(\boldsymbol{V}\boldsymbol{S}^\mathsf{T}\boldsymbol{S}\boldsymbol{V}^\mathsf{T}\sigma_{\mathsf{y}}^2/\widehat{\sigma}_{\mathsf{y}}^4 + \boldsymbol{I}/\nu)(\boldsymbol{V}\boldsymbol{S}^\mathsf{T}\boldsymbol{S}\boldsymbol{V}^\mathsf{T}/\widehat{\sigma}_{\mathsf{y}}^2 + \boldsymbol{I}/\nu)^{-1} \tag{61}$$

$$= \boldsymbol{V}\operatorname{Diag}(\boldsymbol{\gamma})\boldsymbol{V}^\mathsf{T} \text{ for } \gamma_i = \frac{s_i^2\sigma_{\mathsf{y}}^2/\widehat{\sigma}_{\mathsf{y}}^4 + 1/\nu}{[s_i^2/\widehat{\sigma}_{\mathsf{y}}^2 + 1/\nu]^2}. \tag{62}$$

The desired renoising variance then becomes

$$\boldsymbol{\Sigma} = \sigma^2\boldsymbol{I} - \boldsymbol{C} = \boldsymbol{V}\operatorname{Diag}(\boldsymbol{\lambda})\boldsymbol{V}^\mathsf{T} \text{ for } \lambda_i = \sigma^2 - \frac{s_i^2\sigma_{\mathsf{y}}^2/\widehat{\sigma}_{\mathsf{y}}^4 + 1/\nu}{[s_i^2/\widehat{\sigma}_{\mathsf{y}}^2 + 1/\nu]^2} \tag{63}$$

and we can generate the colored noise $\boldsymbol{c}$ via (16) if the SVD is practical. If not, we approximate $\boldsymbol{\Sigma}$ by

$$\widehat{\boldsymbol{\Sigma}} = (\sigma^2 - \nu)\boldsymbol{I} + \xi\boldsymbol{A}^\mathsf{T}\boldsymbol{A} \text{ with } \xi = \frac{1}{s_{\mathsf{max}}^2}\left(\nu - \frac{s_{\mathsf{max}}^2\sigma_{\mathsf{y}}^2/\widehat{\sigma}_{\mathsf{y}}^4 + 1/\nu}{[s_{\mathsf{max}}^2/\widehat{\sigma}_{\mathsf{y}}^2 + 1/\nu]^2}\right) \tag{64}$$

and generate the colored noise $\boldsymbol{c}$ via (19). It is straightforward to show that $\xi \geq 0$ whenever $\widehat{\sigma}_{\mathsf{y}} \geq \sigma_{\mathsf{y}}$, in which case $\sigma^2 \geq \nu$ guarantees that $\widehat{\boldsymbol{\Sigma}}$ is a valid covariance matrix. Figure 6 shows the close agreement between the ideal and approximate $\widehat{\boldsymbol{\Sigma}}$-renoised error spectra both when $\widehat{\sigma}_{\mathsf{y}} = \sigma_{\mathsf{y}}$ and when $\widehat{\sigma}_{\mathsf{y}} > \sigma_{\mathsf{y}}$.

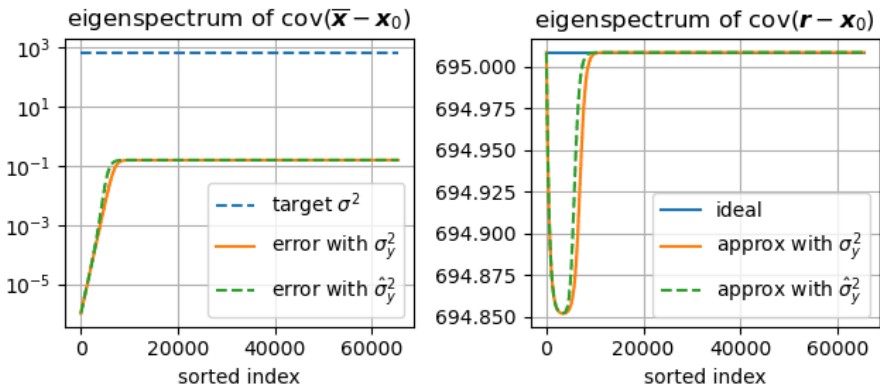

Figure 6: For FFHQ Gaussian deblurring, the left plot shows the eigenspectrum of the error covariance $\text{Cov}\{\overline{\boldsymbol{x}} - \boldsymbol{x}_0\}$ with either $\widehat{\sigma}_{\mathsf{y}}^2$ from (56) (if CG speedup) or $\widehat{\sigma}_{\mathsf{y}}^2 = \sigma_{\mathsf{y}}^2$ (if no CG speedup), as well as the eigenspectrum of the target error covariance $\sigma^2 \boldsymbol{I}$ to aim for when renoising. The right plot shows the eigenvalues of the renoised error covariance $\text{Cov}\{\boldsymbol{r} - \boldsymbol{x}_0\}$ for the ideal case when $\boldsymbol{\Sigma}$ is used (possible with SVD) and the case when $\widehat{\boldsymbol{\Sigma}}$ from (17) is used (if no SVD), with either $\widehat{\sigma}_{\mathsf{y}}^2$ or $\sigma_{\mathsf{y}}^2$. Here we used $\sigma_{\mathsf{y}}^2 = 10^{-6}$, $\nu = 0.16$ (corresponding to the first FIRE iteration of the first DDIM step), and $\rho = 35.7$ (corresponding to the example in Fig. 3).

## E    Proof of Theorem 1

To prove Theorem 1, we begin by writing the key FIRE steps with explicit iteration index $n \geq 1$:

$$\overline{\boldsymbol{x}}[n] = \boldsymbol{d}_{\boldsymbol{\theta}}(\boldsymbol{r}[n], \sigma[n]) \tag{65}$$

$$\widehat{\boldsymbol{x}}[n] = \left(\boldsymbol{A}^{\mathsf{T}}\boldsymbol{A} + \frac{\sigma_{\mathsf{y}}^2}{\nu[n]}\boldsymbol{I}\right)^{-1}\left(\boldsymbol{A}^{\mathsf{T}}\boldsymbol{y} + \frac{\sigma_{\mathsf{y}}^2}{\nu[n]}\overline{\boldsymbol{x}}[n]\right) \tag{66}$$

$$\sigma^2[n+1] = \max\{\sigma^2[n]/\rho, \nu[n]\} \tag{67}$$

$$\lambda_i[n] = \sigma^2[n+1] - (s_i^2/\sigma_{\mathsf{y}}^2 + 1/\nu[n])^{-1}, \quad i = 1, \ldots, d \tag{68}$$

$$\boldsymbol{n}[n] = \boldsymbol{V}\operatorname{Diag}(\boldsymbol{\lambda}[n])^{1/2}\boldsymbol{\varepsilon}[n], \quad \boldsymbol{\varepsilon}[n] \sim \mathcal{N}(\boldsymbol{0}, \boldsymbol{I}) \tag{69}$$

$$\boldsymbol{r}[n+1] = \widehat{\boldsymbol{x}}[n] + \boldsymbol{n}[n] \tag{70}$$

Our proof uses induction. By the assumptions of the theorem, we know that there exists an iteration $n$ (in particular $n = 1$) for which $\boldsymbol{r}[n] = \boldsymbol{x}_0 + \sigma[n]\boldsymbol{\epsilon}[n]$ with $\boldsymbol{\epsilon}[n] \sim \mathcal{N}(\boldsymbol{0}, \boldsymbol{I})$ and finite $\sigma[n]$. Then due to the denoiser assumption, we know that $\overline{\boldsymbol{x}}[n] = \boldsymbol{x}_0 - \sqrt{\nu[n]}\boldsymbol{e}[n]$ with $\boldsymbol{e}[n] = \mathcal{N}(\boldsymbol{0}, \boldsymbol{I})$ and known $\nu[n] < \sigma^2[n]$. We assume that this value of $\nu[n]$ is used in lines (66)-(68). Using these results and (6), we can rewrite (66) as

$$\widehat{\boldsymbol{x}}[n] = \left(\boldsymbol{A}^{\mathsf{T}}\boldsymbol{A} + \frac{\sigma_{\mathsf{y}}^2}{\nu[n]}\boldsymbol{I}\right)^{-1}\left(\boldsymbol{A}^{\mathsf{T}}(\boldsymbol{A}\boldsymbol{x}_0 + \sigma_{\mathsf{y}}\boldsymbol{w}) + \frac{\sigma_{\mathsf{y}}^2}{\nu[n]}(\boldsymbol{x}_0 - \sqrt{\nu[n]}\boldsymbol{e}[n])\right) \tag{71}$$

$$= \boldsymbol{x}_0 + \left(\boldsymbol{A}^{\mathsf{T}}\boldsymbol{A} + \frac{\sigma_{\mathsf{y}}^2}{\nu[n]}\boldsymbol{I}\right)^{-1}\left(\sigma_{\mathsf{y}}\boldsymbol{A}^{\mathsf{T}}\boldsymbol{w} - \frac{\sigma_{\mathsf{y}}^2}{\sqrt{\nu[n]}}\boldsymbol{e}[n]\right) \sim \mathcal{N}(\boldsymbol{x}_0, \boldsymbol{C}[n]) \tag{72}$$

for

$$\boldsymbol{C}[n] \triangleq \left(\boldsymbol{A}^{\mathsf{T}}\boldsymbol{A} + \frac{\sigma_{\mathsf{y}}^2}{\nu[n]}\boldsymbol{I}\right)^{-1}\left(\sigma_{\mathsf{y}}^2\boldsymbol{A}^{\mathsf{T}}\boldsymbol{A} + \frac{\sigma_{\mathsf{y}}^4}{\nu[n]}\boldsymbol{I}\right)\left(\boldsymbol{A}^{\mathsf{T}}\boldsymbol{A} + \frac{\sigma_{\mathsf{y}}^2}{\nu[n]}\boldsymbol{I}\right)^{-1} = \left(\frac{1}{\sigma_{\mathsf{y}}^2}\boldsymbol{A}^{\mathsf{T}}\boldsymbol{A} + \frac{1}{\nu[n]}\boldsymbol{I}\right)^{-1} \tag{73}$$

by leveraging the independent-Gaussian assumption on $\boldsymbol{e}[n]$. From this and (68)-(69), we can then deduce

$$\boldsymbol{n}[n] \sim \mathcal{N}(\boldsymbol{0}, \boldsymbol{\Sigma}[n]) \text{ with } \boldsymbol{\Sigma}[n] \triangleq \boldsymbol{V}\operatorname{Diag}(\boldsymbol{\lambda}[n])\boldsymbol{V}^{\mathsf{T}} = \sigma^2[n+1]\boldsymbol{I} - \boldsymbol{C}[n] \tag{74}$$

---

**Algorithm 3** FIRE for the GLM: $\widehat{\boldsymbol{x}} = \mathsf{FIRE}_{\mathsf{GLM}}(\boldsymbol{y}, \boldsymbol{A}, p_{\mathsf{y|z}}, \boldsymbol{r}_{\mathsf{init}}, \sigma_{\mathsf{init}}, N, \rho)$.

---

**Require:** $\boldsymbol{y}, \boldsymbol{A}, s_{\mathsf{max}}, p_{\mathsf{y|z}}, N, \rho > 1, \boldsymbol{r}_{\mathsf{init}}, \sigma_{\mathsf{init}}$. Also $\boldsymbol{A} = \boldsymbol{U} \operatorname{Diag}(\boldsymbol{s}) \boldsymbol{V}^{\mathsf{T}}$ if using SVD.

1:    $\boldsymbol{r} = \boldsymbol{r}_{\mathsf{init}}$ and $\sigma = \sigma_{\mathsf{init}}$          ▷ *Initialize*

2: **for** $n = 1, \dots, N$ **do**

3:     $\overline{\boldsymbol{x}} \leftarrow \boldsymbol{d_\theta}(\boldsymbol{r}, \sigma) + \sqrt{\widehat{\nu}_\phi(\sigma)}\boldsymbol{v}, \quad \boldsymbol{v} \sim \mathcal{N}(\boldsymbol{0}, \boldsymbol{I})$       ▷ *Stochastic denoising*

4:     $\nu \leftarrow 2\widehat{\nu}_\phi(\sigma)$       ▷ *Error variance of* $\overline{\boldsymbol{x}}$

5:     $\overline{\boldsymbol{z}} \leftarrow \boldsymbol{A}\overline{\boldsymbol{x}}$       ▷ *Denoised version of* $\boldsymbol{A}\boldsymbol{x}_0$

6:     $\overline{\nu}_{\mathsf{z}} \leftarrow \nu \|\boldsymbol{A}\|_F^2/m$       ▷ *Error variance of* $\overline{\boldsymbol{z}}$

7:     $\widehat{z}_j \leftarrow \mathrm{E}\{z_{0,j}|y_j; \overline{z}_j, \overline{\nu}_{\mathsf{z}}\} \; \forall j = 1, \dots, m$       ▷ *Estimate* $z_{0,j} \sim \mathcal{N}(\overline{z}_j, \overline{\nu}_{\mathsf{z}})$ *from* $y_j \sim p_{\mathsf{y|z}}(\cdot|z_{0,j})$

8:     $\widehat{\nu}_{\mathsf{z}} \leftarrow \frac{1}{m}\sum_{j=1}^m \mathrm{var}\{z_{0,j}|y_j; \overline{z}_j, \overline{\nu}_{\mathsf{z}}\}$       ▷ *Averaged posterior variance of* $\{z_{0,j}\}$

9:     $\overline{\sigma}_{\mathsf{y}}^2 \leftarrow [1/\widehat{\nu}_{\mathsf{z}} - 1/\overline{\nu}_{\mathsf{z}}]^{-1}$       ▷ *Extrinsic variance*

10:    $\overline{\boldsymbol{y}} \leftarrow \overline{\sigma}_{\mathsf{y}}^2(\widehat{\boldsymbol{z}}/\widehat{\nu}_{\mathsf{z}} - \overline{\boldsymbol{z}}/\overline{\nu}_{\mathsf{z}})$       ▷ *Extrinsic mean*

11:    $\nu \leftarrow (\|\overline{\boldsymbol{y}} - \boldsymbol{A}\overline{\boldsymbol{x}}\|^2 - \overline{\sigma}_{\mathsf{y}}^2 m)/\|\boldsymbol{A}\|_F^2$       ▷ *Error variance of* $\overline{\boldsymbol{x}}$

12:    $\widehat{\boldsymbol{x}} \leftarrow \arg\min_{\boldsymbol{x}} \|\overline{\boldsymbol{y}} - \boldsymbol{A}\boldsymbol{x}\|^2/\overline{\sigma}_{\mathsf{y}}^2 + \|\boldsymbol{x} - \overline{\boldsymbol{x}}\|^2/\nu$       ▷ *Estimate* $\boldsymbol{x}_0 \sim \mathcal{N}(\overline{\boldsymbol{x}}, \nu\boldsymbol{I})$ *from* $\overline{\boldsymbol{y}} \sim \mathcal{N}(\boldsymbol{A}\boldsymbol{x}_0; \overline{\sigma}_{\mathsf{y}}^2\boldsymbol{I})$

13:    $\sigma^2 \leftarrow \max\{\sigma^2/\rho, \nu\}$       ▷ *Decrease target variance*

14:    **if** have SVD **then**

15:      $\lambda_i \leftarrow \sigma^2 - (s_i^2/\sigma_{\mathsf{y}}^2 + 1/\nu)^{-1}, \quad i = 1, \dots, d$

16:      $\boldsymbol{c} \leftarrow \boldsymbol{V} \operatorname{Diag}(\boldsymbol{\lambda})^{1/2}\boldsymbol{\varepsilon}, \quad \boldsymbol{\varepsilon} \sim \mathcal{N}(\boldsymbol{0}, \boldsymbol{I})$       ▷ *Colored Gaussian noise*

17:    **else**

18:      $\xi \leftarrow \big(\nu - (s_{\mathsf{max}}^2/\sigma_{\mathsf{y}}^2 + 1/\nu)^{-1}\big)/s_{\mathsf{max}}^2$

19:      $\boldsymbol{c} \leftarrow \big[\sqrt{\sigma^2 - \nu}\boldsymbol{I} \; \sqrt{\xi}\boldsymbol{A}^{\mathsf{T}}\big]\boldsymbol{\varepsilon}, \quad \boldsymbol{\varepsilon} \sim \mathcal{N}(\boldsymbol{0}, \boldsymbol{I})$       ▷ *Colored Gaussian noise*

20:    $\boldsymbol{r} \leftarrow \widehat{\boldsymbol{x}} + \boldsymbol{c}$       ▷ *Renoise so that* $\mathrm{Cov}\{\boldsymbol{r} - \boldsymbol{x}_0\} = \sigma^2\boldsymbol{I}$

21: **return** $\widehat{\boldsymbol{x}}$

---

so that, from (70),

$$\boldsymbol{r}[n+1] \sim \mathcal{N}(\boldsymbol{x}_0, \boldsymbol{C}[n] + \boldsymbol{\Sigma}[n]) = \mathcal{N}(\boldsymbol{x}_0, \sigma^2[n+1]\boldsymbol{I}) \tag{75}$$

$$\Leftrightarrow \boldsymbol{r}[n+1] = \boldsymbol{x}_0 + \sigma[n+1]\boldsymbol{\epsilon}[n+1], \quad \boldsymbol{\epsilon}[n+1] \sim \mathcal{N}(\boldsymbol{0}, \boldsymbol{I}). \tag{76}$$

Thus, by induction, if $\boldsymbol{r}[n] = \boldsymbol{x}_0 + \sigma[n]\boldsymbol{\epsilon}[n]$ with $\boldsymbol{\epsilon}[n] \sim \mathcal{N}(\boldsymbol{0}, \boldsymbol{I})$ holds at $n = 1$, then it holds at all $n > 1$.

Recall that the theorem also assumed that $\nu[n] < \sigma^2[n]$ for all $n$. Thus, there exists a $\rho > 1$ for which $\sigma^2[n]/\rho > \nu[n]$ for all $n$, for which we can rewrite (67) as

$$\sigma^2[n+1] = \sigma^2[n]/\rho \; \forall n. \tag{77}$$

Consequently, for any iteration $n \geq 1$ we can write

$$\sigma^2[n] = \sigma^2[1]/\rho^{n-1} = \sigma_{\mathsf{init}}^2/\rho^{n-1}. \tag{78}$$

Finally, because the error covariance on $\widehat{\boldsymbol{x}}[n]$ obeys

$$\boldsymbol{C}[n] = \left(\frac{1}{\sigma_{\mathsf{y}}^2}\boldsymbol{A}^{\mathsf{T}}\boldsymbol{A} + \frac{1}{\nu[n]}\boldsymbol{I}\right)^{-1} < \nu[n]\boldsymbol{I} < \sigma^2[n]\boldsymbol{I} = \frac{\sigma_{\mathsf{init}}^2}{\rho^{n-1}}\boldsymbol{I} \tag{79}$$

we see that the error variance in $\widehat{\boldsymbol{x}}[n]$ decreases exponentially with $n$ and thus $\widehat{\boldsymbol{x}}[n]$ converges to the true $\boldsymbol{x}_0$.

## F   GLM-FIRE algorithm

The full algorithm for the GLM version of FIRE is given in Alg. 3. There, the dashed blue box indicates the lines used for the EP update; the remaining lines mirror those in SLM-FIRE, summarized as Alg. 1.

---

**Algorithm 4** DDS (Chung et al., 2024a)

---

**Require:** $d_{\boldsymbol{\theta}}(\cdot, \cdot), \boldsymbol{y}, \boldsymbol{A}, \gamma_{\mathsf{dds}}, M_{\mathsf{cg}}, \{\sigma_k\}_{k=1}^{K}, \eta_{\mathsf{ddim}} \geq 0$

  1: $\boldsymbol{x}_K \sim \mathcal{N}(\boldsymbol{0}, \sigma_K^2 \boldsymbol{I})$
  2: **for** $k = K, K-1, \dots, 1$ **do**
  3:      $\overline{\boldsymbol{x}}_k = d_{\boldsymbol{\theta}}(\boldsymbol{x}_k, \sigma_k)$                                ▷ *Denoising*
  4:      $\widehat{\boldsymbol{x}}_{0|k} = \mathsf{CG}(\boldsymbol{A}^\mathsf{T}\boldsymbol{A} + \gamma_{\mathsf{dds}}\boldsymbol{I}, \boldsymbol{A}^\mathsf{T}\boldsymbol{y} + \gamma_{\mathsf{dds}}\overline{\boldsymbol{x}}_k, \overline{\boldsymbol{x}}_k, M_{\mathsf{cg}}) \approx \arg\min_{\boldsymbol{x}} \left\{ \|\boldsymbol{y} - \boldsymbol{A}\boldsymbol{x}\|^2 + \gamma_{\mathsf{dds}}\|\boldsymbol{x} - \overline{\boldsymbol{x}}_k\|^2 \right\}$
  5:      $\varsigma_k = \eta_{\mathsf{ddim}}\sqrt{\dfrac{\sigma_{k-1}^2(\sigma_k^2 - \sigma_{k-1}^2)}{\sigma_k^2}}$
  6:      $\boldsymbol{x}_{k-1} = \sqrt{\dfrac{\sigma_{k-1}^2 - \varsigma_k^2}{\sigma_k^2}}\boldsymbol{x}_k + \left(1 - \sqrt{\dfrac{\sigma_{k-1}^2 - \varsigma_k^2}{\sigma_k^2}}\right)\widehat{\boldsymbol{x}}_{0|k} + \varsigma_k\boldsymbol{n}_k, \quad \boldsymbol{n}_k \sim \mathcal{N}(\boldsymbol{0}, \boldsymbol{I})$       ▷ *DDIM update*
  7: **return** $\widehat{\boldsymbol{x}}_{0|k}$

---

---

**Algorithm 5** DiffPIR (Zhu et al., 2023)

---

**Require:** $d_{\boldsymbol{\theta}}(\cdot, \cdot), \boldsymbol{y}, \boldsymbol{A}, \sigma_{\mathsf{y}}, \lambda_{\mathsf{diffpir}}, \{\sigma_k\}_{k=1}^{K}, \eta_{\mathsf{diffpir}}$

  1: $\boldsymbol{x}_K \sim \mathcal{N}(\boldsymbol{0}, \sigma_K^2 \boldsymbol{I})$
  2: **for** $k = K, K-1, \dots, 1$ **do**
  3:      $\overline{\boldsymbol{x}}_k = d_{\boldsymbol{\theta}}(\boldsymbol{x}_k, \sigma_k)$                                ▷ *Denoising*
  4:      $\widehat{\boldsymbol{x}}_{0|k} = \arg\min_{\boldsymbol{x}} \left\{ \dfrac{1}{2\sigma_{\mathsf{y}}^2}\|\boldsymbol{y} - \boldsymbol{A}\boldsymbol{x}\|^2 + \dfrac{\lambda_{\mathsf{diffpir}}}{2\sigma_k^2}\|\boldsymbol{x} - \overline{\boldsymbol{x}}_k\|^2 \right\}$       ▷ *Approximate MMSE estimation*
  5:      $\varrho_k = \sqrt{1 - \eta_{\mathsf{diffpir}}}\dfrac{\sigma_{k-1}}{\sigma_k}$
  6:      $\boldsymbol{x}_{k-1} = \varrho_k\boldsymbol{x}_k + (1 - \varrho_k)\widehat{\boldsymbol{x}}_{0|k} + \sqrt{\eta_{\mathsf{diffpir}}}\sigma_{k-1}\boldsymbol{n}_k, \quad \boldsymbol{n}_k \sim \mathcal{N}(\boldsymbol{0}, \boldsymbol{I})$       ▷ *DDIM-like update*
  7: **return** $\widehat{\boldsymbol{x}}_{0|k}$

---

## G  Comparison to other renoising proximal-data-fidelity PnP schemes

In this section, we compare SLM-FIRE/DDfire to other renoising PnP schemes for the SLM that involve a proximal data-fidelity step or, equivalently, MMSE estimation of $\boldsymbol{x}_0$ under the Gaussian prior assumption (8): DDS, DiffPIR, and SNORE, which are detailed in Alg. 4, Alg. 5, and Alg. 6, respectively.

Comparing SLM-FIRE/DDfire (see Alg. 1 and Alg. 2) to DDS, we see that both approaches use CG for approximate MMSE estimation under a Gaussian prior approximation and both use a DDIM diffusion update. But DDS uses the hyperparameters $\gamma_{\mathsf{dds}}$ and $M_{\mathsf{cg}}$ (the number CG iterations, which is usually small, such as four) to adjust the noise variance $\nu$ in the Gaussian prior approximation (8), while SLM-FIRE/DDfire estimates $\nu$ at each iteration. Also, SLM-FIRE/DDfire injects colored Gaussian noise to whiten the denoiser input error while DDS injects no noise outside of DDIM.

Comparing SLM–IRE/DDfire to DiffPIR, we see that both perform MMSE estimation under a Gaussian prior approximation and both use a DDIM-like diffusion update. But DiffPIR uses the hyperparameter $\lambda_{\mathsf{diffpir}}$ to adjust the noise variance $\nu$ in the Gaussian prior approximation (8), while SLM-FIRE/DDfire estimates $\nu$ at each iteration. Also, SLM-FIRE/DDfire injects colored Gaussian noise to whiten the denoiser input error while DiffPIR injects no noise outside of its DDIM-like step.

Comparing SLM-FIRE to SNORE, we see that both perform MMSE estimation under a Gaussian prior approximation and both use renoising. But SNORE uses white renoising while SLM-FIRE uses colored renoising to whiten the denoiser input error. Also, SNORE uses the hyperparameter $\delta_{\mathsf{snore}}$ to adjust the noise variance $\nu$ in the Gaussian prior approximation (8), while SLM-FIRE estimates it at each iteration. Furthermore, since SNORE is based on the RED algorithm (Romano et al., 2017), its denoiser output is scaled and shifted. Finally, SNORE has many more tuning parameters than SLM-FIRE.

---

**Algorithm 6** Annealed Proximal SNORE (Renaud et al., 2024)

---

**Require:** $d_{\boldsymbol{\theta}}(\cdot,\cdot), \boldsymbol{y}, \boldsymbol{A}, \sigma_{\mathsf{y}}, \delta_{\mathsf{snore}}, M_{\mathsf{snore}}, \{\sigma_i\}_{i=1}^{M_{\mathsf{snore}}}, \{\alpha_i\}_{i=1}^{M_{\mathsf{snore}}}, \{K_i\}_{i=1}^{M_{\mathsf{snore}}}, \widehat{\boldsymbol{x}}_{\mathsf{init}}$

1: $\widehat{\boldsymbol{x}}_{0|K_{M_{\mathsf{snore}}}} = \widehat{\boldsymbol{x}}_{\mathsf{init}}$
2: **for** $i = M_{\mathsf{snore}}, M_{\mathsf{snore}} - 1 \ldots, 1$ **do**
3:      **for** $k = K_i, K_i - 1, \ldots, 1$ **do**
4:          $\boldsymbol{r}_k = \widehat{\boldsymbol{x}}_{0|k} + \sigma_i \boldsymbol{n}_{i,k}, \quad \boldsymbol{n}_{i,k} \sim \mathcal{N}(\boldsymbol{0}, \boldsymbol{I})$           $\triangleright$ *Renoising*
5:          $\overline{\boldsymbol{x}}_k = \left(1 - \dfrac{\delta_{\mathsf{snore}}\alpha_i}{\sigma_i^2}\right)\widehat{\boldsymbol{x}}_{0|k} + \dfrac{\delta_{\mathsf{snore}}\alpha_i}{\sigma_i^2}d_{\boldsymbol{\theta}}(\boldsymbol{r}_k, \sigma_i)$      $\triangleright$ *RED update*
6:          $\widehat{\boldsymbol{x}}_{0|k-1} = \arg\min_{\boldsymbol{x}}\left\{\dfrac{1}{2\sigma_{\mathsf{y}}^2}\|\boldsymbol{y} - \boldsymbol{A}\boldsymbol{x}\|^2 + \dfrac{1}{2\delta_{\mathsf{snore}}}\|\boldsymbol{x} - \overline{\boldsymbol{x}}_k\|^2\right\}$    $\triangleright$ *Approximate MMSE estimation*
7: **return** $\widehat{\boldsymbol{x}}_{0|k}$

---

## H  Implementation details

### H.1  Inverse problems

For the linear inverse problems, the measurements were generated as

$$\boldsymbol{y} = \boldsymbol{A}\boldsymbol{x}_0 + \sigma_{\mathsf{y}}\boldsymbol{w}, \quad \boldsymbol{w} \sim \mathcal{N}(\boldsymbol{0}, \boldsymbol{I}) \tag{80}$$

with appropriate $\boldsymbol{A}$. For box inpainting, Gaussian deblurring, and super-resolution we used the $\boldsymbol{A}$ and $\boldsymbol{A}^{\mathsf{T}}$ implementations from Kawar et al. (2022b). For motion deblurring, we implemented our own $\boldsymbol{A}$ and $\boldsymbol{A}^{\mathsf{T}}$ with reflect padding. All methods used these operators implementations except DiffPIR, which used the authors' implementations. Motion-blur kernels were generated using Borodenko (2020).

For phase retrieval, the measurements were generated using the method from Metzler et al. (2018):

$$y_j^2 = |z_{0,j}|^2 + w_j, \quad w_j \sim \mathcal{N}(0, \alpha_{\mathsf{shot}}^2|z_{0,j}|^2), \quad j = 1, \ldots, m, \tag{81}$$

where $\alpha_{\mathsf{shot}}$ controls the noise level and $\boldsymbol{z}_0 = \boldsymbol{A}\boldsymbol{x}_0$, with the values of $\boldsymbol{x}_0$ scaled to lie in the range $[0, 255]$. This is an approximation of the Poisson shot-noise corruption model in that the intensity $y_j^2/\alpha_{\mathsf{shot}}^2$ is approximately $\mathrm{Poisson}((|z_{0,j}|/\alpha_{\mathsf{shot}})^2)$ distributed for sufficiently small values of $\alpha_{\mathsf{shot}}$. We implemented the oversampled-Fourier $\boldsymbol{A}$ by zero-padding the image by $2\times$ in each direction and then passing the result through a unitary FFT. For CDP phase retrieval, we set $\boldsymbol{A} = [\boldsymbol{A}_1^{\mathsf{T}}, \ldots, \boldsymbol{A}_L^{\mathsf{T}}]^{\mathsf{T}}$ for $\boldsymbol{A}_l = L^{-1/2}\boldsymbol{F}\mathrm{Diag}(\boldsymbol{c}_l)$, where $\boldsymbol{F}$ is a $d \times d$ FFT and $\boldsymbol{c}_l$ contain i.i.d. random entries uniformly distributed on the unit circle in the complex plane, and where $L = 4$. In both cases, $\boldsymbol{A}^{\mathsf{T}}\boldsymbol{A} = \boldsymbol{I}$.

### H.2  Evaluation protocol

For the linear inverse problems, we run each method once for each measurement $\boldsymbol{y}$ in the 1000-sample test set and compute average PSNR, average LPIPS, and FID from the resulting recoveries.

For OSF phase retrieval, following Chung et al. (2023a), we run each algorithm four times and keep the reconstruction $\widehat{\boldsymbol{x}}$ that minimizes the measurement residual $\|\boldsymbol{y} - |\boldsymbol{A}\widehat{\boldsymbol{x}}|\|$. Performance metrics are then evaluated after resolving the inherent spatial shift and conjugate flip ambiguities associated with phase retrieval (see, e.g., Bendory et al. (2015)). Note global phase ambiguity is not an issue due to the non-negativity of our images. For the CDP experiments, we run each algorithm only once and don't perform ambiguity resolution, because it is unnecessary.

### H.3  Unconditional diffusion models

For the FFHQ experiments, all methods used the pretrained model from Chung et al. (2023a). For the ImageNet experiments, all methods used the pretrained model from Dhariwal & Nichol (2021). In both cases, $T = 1000$.

Table 5: Hyperparameter values used for DDfire.

| | | Inpaint (box) | | Deblur (Gaussian) | | Deblur (Motion) | | $4\times$ Super-resolution | |
|---|---|---|---|---|---|---|---|---|---|
| Dataset | $\sigma_{\mathsf{y}}$ | $K$ | $\delta$ | $K$ | $\delta$ | $K$ | $\delta$ | $K$ | $\delta$ |
| FFHQ | 0.05 | 100 | 0.50 | 650 | 0.60 | 500 | 0.20 | 650 | 0.60 |
| ImageNet | 0.05 | 100 | 0.50 | 500 | 0.20 | 500 | 0.20 | 650 | 0.60 |

Table 6: Hyperparameter values used for DDfire phase retrieval.

| Operator | $\alpha_{\mathsf{shot}}$ | $K$ | $\delta$ |
|---|---|---|---|
| OSF | 8 | 20 | 0.00 |
| CDP | 45 | 10 | 0.00 |

### H.4 Recovery methods

**DDfire.** Our Python/Pytorch codebase is a modification of the DPS codebase from Chung et al. (2023b) and is available at `https://github.com/matt-bendel/DDfire`. For all but one row of the ablation study in Table 1 and the dashed line in Fig. 4, we ran DDfire without an SVD and thus with the approximate renoising in (19).

For the linear inverse problems, unless noted otherwise, we ran DDfire for 1000 NFEs using $\eta_{\mathsf{ddim}} = 1.5$, and we did not use stochastic denoising (i.e., $\widehat{\nu}_\phi(\sigma) = 0 \; \forall \sigma$ in Alg. 1), as suggested by our ablation study. We tuned the $(K, \delta)$ hyperparameters to minimize LPIPS on a 100-sample validation set, yielding the parameters in Table 5. For the runtime results in Fig. 4, we used $\eta_{\mathsf{ddim}} = 1.0$ for $N_{\mathsf{tot}} \in \{50, 100, 200, 500\}$ and $\eta_{\mathsf{ddim}} = 0$ for $N_{\mathsf{tot}} = 20$, and we used $K = \text{NFE}/2$ and $\delta = 0.2$ for all cases. For the $\nu$-estimation step in Alg. 1, we used $\|\boldsymbol{A}\|_F^2 \approx \frac{1}{L} \sum_{l=1}^{L} \|\boldsymbol{A}\boldsymbol{w}_l\|^2$ with i.i.d. $\boldsymbol{w}_l \sim \mathcal{N}(\boldsymbol{0}, \boldsymbol{I})$ and $L = 25$.

For phase retrieval, we ran DDfire using a default of 800 NFEs for OSF and 100 NFEs for CDP. We set $\eta_{\mathsf{ddim}} = 0.85$ and the hand-tuned $(K, \delta)$ values listed in Table 6. Also, we did use stochastic denoising, since it helped in all metrics. Since the likelihood $p_{\mathsf{y}|\mathsf{z}}(y|z) = \mathcal{N}(y; |z|, \sigma_{\mathsf{y}}^2)$ makes the conditional mean and variance in lines 8-7 of Alg. 3 intractable, we used the Laplace approximation (Bishop, 2007). For the special case of ImageNet OSF, we guided DDfire using the HIO estimate; in all other cases, DDfire received no guidance. Appendix H.5 details the guidance.

**DDRM.** We ran DDRM for 20 NFEs using the authors' implementation from Kawar et al. (2022b) with minor changes to work with our codebase.

**DiffPIR.** We ran DiffPIR for 20 NFEs using the authors' implementation from Zhu et al. (2024) without modification. Hyperparameters were set according to the reported values in Zhu et al. (2023).

**ΠGDM.** We ran ΠGDM for 100 NFEs. Since the authors do not provide a ΠGDM implementation for noisy inverse problems in NVlabs (2023), we coded ΠGDM ourselves in Python/PyTorch. With problems for which an SVD is available, we computed $(\boldsymbol{A}\boldsymbol{A}^\mathsf{T} + \zeta_k \boldsymbol{I})^{-1}$ using the efficient SVD implementation of $\boldsymbol{A}$ from the DDRM codebase Kawar et al. (2022b), and otherwise we used CG.

**DDS.** We ran DDS for 100 NFEs. We leveraged the authors' implementation in Chung et al. (2024b) to reimplement DDS in our codebase. We tuned the DDS regularization parameter $\gamma_{\mathsf{dds}}$ via grid search and used $\eta_{\mathsf{ddim}} = 0.8$ and 50 CG iterations.

**DPS.** For the linear inverse problems, we ran DPS for 1000 NFEs using the authors' implementation from Chung et al. (2023b) without modification, using the suggested tuning from Chung et al. (2023a, Sec. D.1).

For phase retrieval, we also used 1000 NFEs, but made minor adjustments to the DPS authors' implementation to accommodate the likelihood $p_{\mathsf{y}|\mathsf{z}}(y|z) = \mathcal{N}(y; |z|, \sigma_{\mathsf{y}}^2)$, which was used by all methods for fairness. We used grid-search tuning to minimize LPIPS on a 100-image validation set.

**RED-diff.** We ran RED-diff for 1000 NFEs using the authors' implementation from NVlabs (2023), with minor changes to work with our codebase. We tuned the RED-diff learning rate, $\lambda$, and data fidelity weight $v_t$ to minimize LPIPS with a 100-image validation set. For phase retrieval, we made similar code modifications to handle our likelihood as with DPS.

**DAPS.** We ran DAPS for 1000 NFEs using the authors' implementation from Zhang et al. (2024), with minor changes to work in our codebase. The tuning parameters were set as in Zhang et al. (2025). For phase retrieval, we made similar code modifications to handle our likelihood as with DPS.

**DOLPH.** We ran DOLPH for 1000 NFEs. Since the DOLPH authors did not release an implementation, we implemented it ourselves in Python/PyTorch and used grid-search to find the step-size that minimized LPIPS on a 100-image validation set. For phase retrieval, we made similar code modifications to handle our likelihood as with DPS.

**HIO.** We translated the MATLAB implementation of HIO from Metzler (2018) to Python and set the step-size parameter to 0.9. We then followed the runtime procedure described in Metzler et al. (2018): For the OSF experiments, HIO is first run 50 times, for 50 iterations each, from a random initialization. The estimate $\widehat{\boldsymbol{x}}$ with the lowest measurement residual $\|\boldsymbol{y} - |\boldsymbol{A}\widehat{\boldsymbol{x}}|\|$ is then used to reinitialize HIO, after which it is run for 1000 more iterations. Finally, the second and third color channels in the result are shifted and flipped as needed to best match the first color channel. For the CDP experiments, HIO is run once for 200 iterations from a random initialization.

**prDeep.** We used the Python implementation from Hekstra et al. (2018). As recommended in Metzler et al. (2018), we initialized prDeep with the HIO estimate for OSF experiments and with an all-ones initialization for CDP experiments (The HIO estimate was also used by DDfire for ImageNet OSF). We tuned $\lambda$ on a grid to minimize LPIPS on a 100-image validation set.

### H.5 DDfire with guidance

Algorithm 7 details an extension of DDfire that takes guidance from an externally provided estimate $\widehat{\boldsymbol{x}}_{\mathsf{guide}}$. The approach can be explained as follows. Recall from (38) and App. C that the iteration-$k$ VE DDIM estimate $\boldsymbol{x}_k$ is assumed to obey the model

$$p(\boldsymbol{x}_k|\boldsymbol{x}_0) = \mathcal{N}(\boldsymbol{x}_k; \boldsymbol{x}_0, \sigma_k^2 \boldsymbol{I}). \tag{82}$$

If we set $\widetilde{\boldsymbol{x}}_{\mathsf{guide}} = \widehat{\boldsymbol{x}}_{\mathsf{guide}} + \sigma_{\mathsf{guide}} \boldsymbol{v}_k$ with $\boldsymbol{v}_k \sim \mathcal{N}(\boldsymbol{0}, \boldsymbol{I})$ and sufficiently large $\sigma_{\mathsf{guide}}$, then it is reasonable to model

$$p(\widetilde{\boldsymbol{x}}_{\mathsf{guide}}|\boldsymbol{x}_0) = \mathcal{N}(\widetilde{\boldsymbol{x}}_{\mathsf{guide}}; \boldsymbol{x}_0, \sigma_{\mathsf{guide}}^2 \boldsymbol{I}). \tag{83}$$

Furthermore, if $\boldsymbol{v}_k$ is generated independently of $\boldsymbol{x}_k$ at each iteration $k$, then

$$p(\boldsymbol{x}_k, \widetilde{\boldsymbol{x}}_{\mathsf{guide}}|\boldsymbol{x}_0) = \mathcal{N}(\boldsymbol{x}_k; \boldsymbol{x}_0, \sigma_k^2 \boldsymbol{I}) \mathcal{N}(\widetilde{\boldsymbol{x}}_{\mathsf{guide}}; \boldsymbol{x}_0, \sigma_{\mathsf{guide}}^2 \boldsymbol{I}) \tag{84}$$

$$= \mathcal{N}(\boldsymbol{x}_0; \boldsymbol{x}_k, \sigma_k^2 \boldsymbol{I}) \mathcal{N}(\boldsymbol{x}_0; \widetilde{\boldsymbol{x}}_{\mathsf{guide}}, \sigma_{\mathsf{guide}}^2 \boldsymbol{I}) \tag{85}$$

$$\propto \mathcal{N}(\boldsymbol{x}_0; \boldsymbol{r}_k, \nu_k \boldsymbol{I}) \;\; \text{for} \;\; \begin{cases} \boldsymbol{r}_k = \dfrac{\sigma_{\mathsf{guide}}^2}{\sigma_k^2 + \sigma_{\mathsf{guide}}^2} \boldsymbol{x}_k + \dfrac{\sigma_k^2}{\sigma_k^2 + \sigma_{\mathsf{guide}}^2} \widetilde{\boldsymbol{x}}_{\mathsf{guide}} \\ \nu_k = \left(\dfrac{1}{\sigma_k^2} + \dfrac{1}{\sigma_{\mathsf{guide}}^2}\right)^{-1} \end{cases} \tag{86}$$

$$= \mathcal{N}(\boldsymbol{r}_k; \boldsymbol{x}_0, \nu_k \boldsymbol{I}), \tag{87}$$

---

**Algorithm 7** DDfire with guidance

---

**Require:** $\boldsymbol{y}, \boldsymbol{A}, \sigma_\mathsf{y}$ or $p_{\mathsf{y}|\mathsf{z}}, \rho, \{\sigma_k\}_{k=1}^K, \{N_k\}_{k=1}^K, \eta_\mathsf{ddim} \geq 0, \widehat{\boldsymbol{x}}_\mathsf{guide}, \sigma_\mathsf{guide}$

1: $\boldsymbol{x}_K \sim \mathcal{N}(\boldsymbol{0}, \sigma_K^2 \boldsymbol{I})$

2: **for** $k = K, K-1, \ldots, 1$ **do**

3: $\quad \widetilde{\boldsymbol{x}}_\mathsf{guide} = \widehat{\boldsymbol{x}}_\mathsf{guide} + \sigma_\mathsf{guide} \boldsymbol{v}_k, \quad \boldsymbol{v}_k \sim \mathcal{N}(\boldsymbol{0}, \boldsymbol{I})$ ▷ *Randomized guidance*

4: $\quad \boldsymbol{r}_k = \dfrac{\sigma_\mathsf{guide}^2}{\sigma_k^2 + \sigma_\mathsf{guide}^2} \boldsymbol{x}_k + \dfrac{\sigma_k^2}{\sigma_k^2 + \sigma_\mathsf{guide}^2} \widetilde{\boldsymbol{x}}_\mathsf{guide}$ ▷ *Guided estimate*

5: $\quad \nu_k = \left( \dfrac{1}{\sigma_k^2} + \dfrac{1}{\sigma_\mathsf{guide}^2} \right)^{-1}$ ▷ *Approximate error-variance of $\boldsymbol{r}_k$*

6: $\quad \widehat{\boldsymbol{x}}_{0|k} = \mathsf{FIRE}(\boldsymbol{y}, \boldsymbol{A}, *, \boldsymbol{r}_k, \sqrt{\nu_k}, N_k, \rho)$ ▷ *FIRE via Alg. 1 or Alg. 3*

7: $\quad \varsigma_k = \eta_\mathsf{ddim} \sqrt{\dfrac{\sigma_{k-1}^2 (\sigma_k^2 - \sigma_{k-1}^2)}{\sigma_k^2}}$

8: $\quad \boldsymbol{x}_{k-1} = \sqrt{\dfrac{\sigma_{k-1}^2 - \varsigma_k^2}{\sigma_k^2}} \boldsymbol{x}_k + \left( 1 - \sqrt{\dfrac{\sigma_{k-1}^2 - \varsigma_k^2}{\sigma_k^2}} \right) \widehat{\boldsymbol{x}}_{0|k} + \varsigma_k \boldsymbol{n}_k, \quad \boldsymbol{n}_k \sim \mathcal{N}(\boldsymbol{0}, \boldsymbol{I})$ ▷ *DDIM update*

9: **return** $\widehat{\boldsymbol{x}}_{0|k}$

---

which means that i) $\boldsymbol{r}_k$ is a sufficient statistic for the estimation of $\boldsymbol{x}_0$ from $\boldsymbol{x}_k$ and $\widetilde{\boldsymbol{x}}_\mathsf{guide}$ (Poor, 1994) and ii) that $\boldsymbol{r}_k$ can be modeled as an AWGN-corrupted version of $\boldsymbol{x}_0$ with AWGN variance $\nu_k$. Thus, whereas standard DDfire passes $(\boldsymbol{x}_k, \sigma_k^2)$ to FIRE in iteration $k$, the guided version of DDfire instead passes $(\boldsymbol{r}_k, \nu_k)$ to FIRE at iteration $k$. Note that this necessitates that $\sigma_k^2$ be replaced by $\nu_k$ when computing $\underline{N}_k$ in (29). Thus concludes the explanation of Alg. 7. As for the choice of $\sigma_\mathsf{guide}^2$, it should be large enough relative to the error variance in $\widehat{\boldsymbol{x}}_\mathsf{guide}$ for (83) to hold. Thus we suggest choosing $\sigma_\mathsf{guide}^2$ to be at least $10\times$ the error variance in $\widehat{\boldsymbol{x}}_\mathsf{guide}$, which can be estimated using (10). But as $\sigma_\mathsf{guide}^2$ grows larger, less guidance is provided. For example, if $\sigma_\mathsf{guide}^2$ is large relative to $\sigma_K^2$, then essentially no guidance will be provided (since $\boldsymbol{r}_k \approx \boldsymbol{x}_k$ and $\nu_k \approx \sigma_k^2$ for all $k \in \{1, \ldots, K\}$ when $\sigma_\mathsf{guide}^2 \gg \sigma_K^2$). For ImageNet OSF phase retrieval, we set $\sigma_\mathsf{guide}^2$ at $50\times$ the error variance of $\widehat{\boldsymbol{x}}_\mathsf{guide} = \widehat{\boldsymbol{x}}_\mathsf{HIO}$.

### H.6 Compute

All experiments were run on a single NVIDIA A100 GPU with 80GB of memory. The runtime for each method on the GPU varies, as shown in Figure 4.

## I DDfire hyperparameter tuning curves

Figures 7–10 show PSNR and LPIPS over the parameter grids $K \in \{10, 20, 50, 100, 200, 500, 1000\}$ and $\delta \in \{0.05, 0.1, 0.2, 0.5, 0.75\}$ for box inpainting, Gaussian deblurring, motion deblurring, and 4x super resolution, respectively, using 50 ImageNet validation images. While there are some noticeable trends, DDfire is relatively sensitive to hyperparemeter selection, particularly in cases where the degradation is not global (e.g., in inpainting).

## J Additional experimental results

Figure 11 shows LPIPS vs. average runtime (in seconds on an A100 GPU) to generate a single image for noisy OSF phase retrieval on the 1000-sample FFHQ test set. The figure shows that DDfire gives a significantly better performance/complexity tradeoff than all diffusion-based competitors.

Figure 12 shows SLM-FIRE's $\sigma^2$ versus iteration $i$, for comparison to the true denoiser input variance $\|\boldsymbol{r} - \boldsymbol{x}_0\|_2^2 / d$, and SLM-FIRE's $\nu$, for comparison to the true denoiser output variance $\|\overline{\boldsymbol{x}}_0 - \boldsymbol{x}_0\|_2^2 / d$, for 25 FIRE iterations with $\rho = 1.5$ for noisy $4\times$ super-resolution at $t[k] = 1000$. We see that the SLM-FIRE estimates $\sigma^2$ and $\nu$ track the true error variances quite closely.

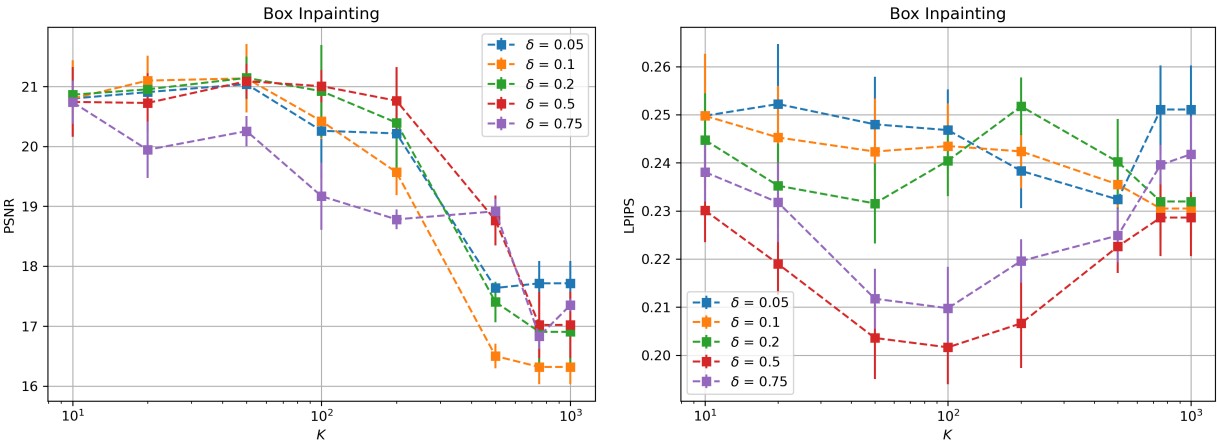

Figure 7: PSNR and LPIPS tuning results for box inpainting with 50 ImageNet validation images.

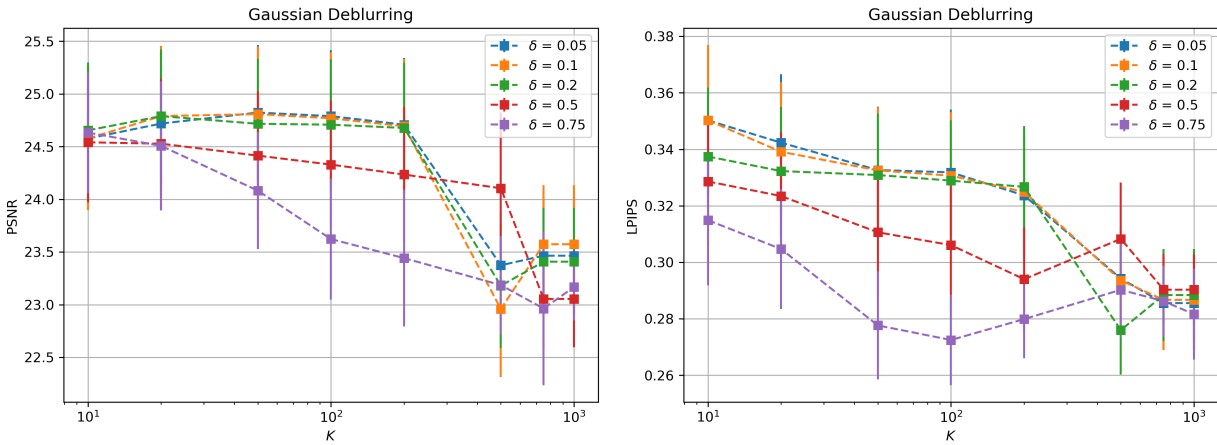

Figure 8: PSNR and LPIPS tuning results for gaussian deblurring with 50 ImageNet validation images.

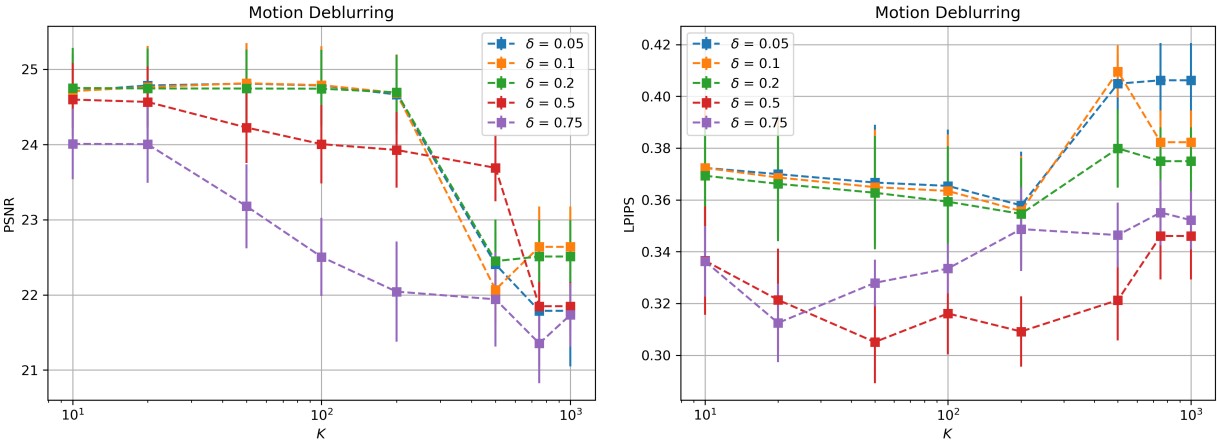

Figure 9: PSNR and LPIPS tuning results for motion deblurring with 50 ImageNet validation images.

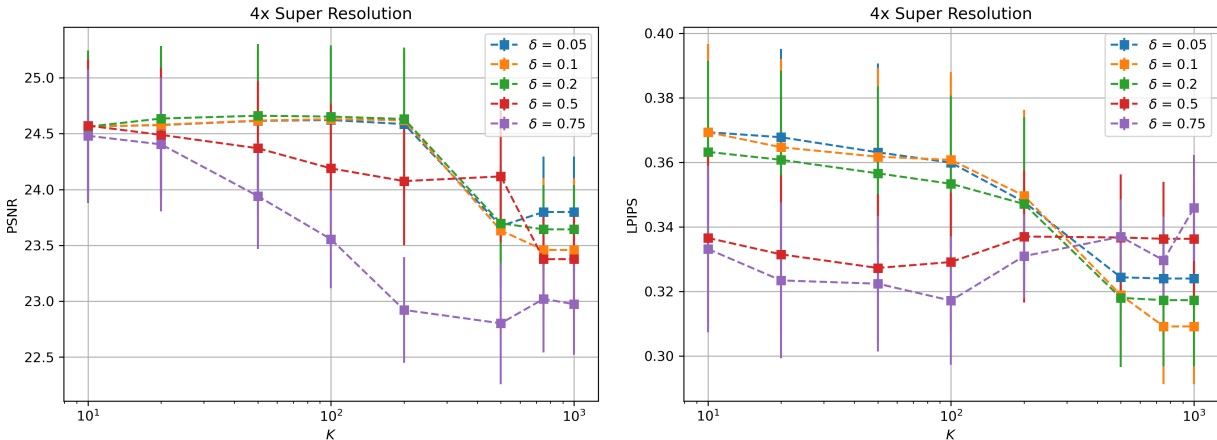

Figure 10: PSNR and LPIPS tuning results for 4x super resolution with 50 ImageNet validation images.

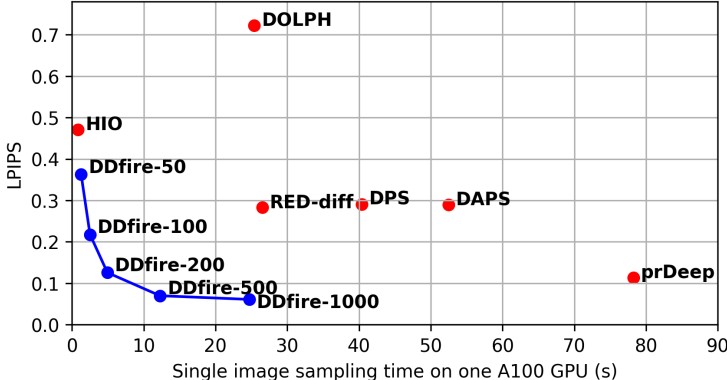

Figure 11: LPIPS vs. single image sampling time for noisy OSF phase retrieval on an A100 GPU. The evaluation used 1000 FFHQ images.

Figure 13 shows a similar figure for GLM-FIRE. In particular, it shows GLM-FIRE's $\sigma^2$ versus iteration $i$, for comparison to the true denoiser input variance $\|\boldsymbol{r} - \boldsymbol{x}_0\|_2^2/d$, GLM-FIRE's $\nu$, for comparison to the true denoiser output variance $\|\overline{\boldsymbol{x}}_0 - \boldsymbol{x}_0\|_2^2/d$, and GLM-FIRE's $\overline{\sigma}_{\mathsf{y}}^2$, for comparison to the true pseudo-measurement variance $\|\overline{\boldsymbol{y}} - \boldsymbol{A}\boldsymbol{x}_0\|^2/m$ for noisy FFHQ phase retrieval at $t[k] = 1000$. We see that the GLM-FIRE estimates $\sigma^2$, $\nu$, and $\overline{\sigma}_{\mathsf{y}}^2$ track the true variances quite closely.

Example recoveries for noisy phase retrieval with FFHQ images are given in Figure 14.

Additional example recoveries for the noisy linear inverse problems with FFHQ images are shown in Figure 15.

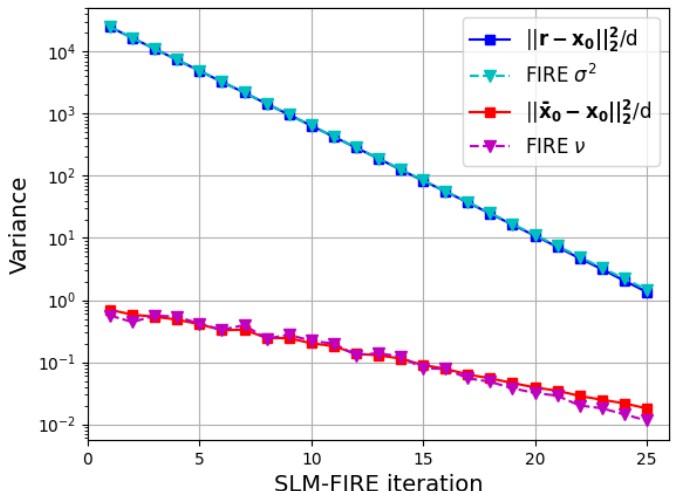

Figure 12: SLM-FIRE $\sigma^2$, true denoiser input variance $\|\boldsymbol{r} - \boldsymbol{x}_0\|_2^2/d$, SLM-FIRE $\nu$, and true denoiser output variance $\|\overline{\boldsymbol{x}}_0 - \boldsymbol{x}_0\|_2^2/d$ vs. SLM-FIRE iteration for noisy $4\times$ super-resolution at $t[k] = 1000$ for a single validation sample $\boldsymbol{x}_0$.

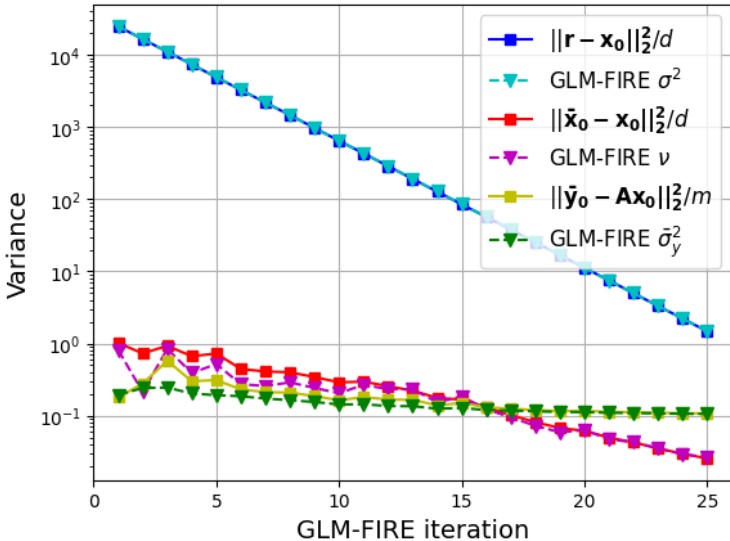

Figure 13: GLM-FIRE $\sigma^2$, true denoiser input variance $\|\boldsymbol{r} - \boldsymbol{x}_0\|_2^2/d$, GLM-FIRE $\nu$, and true denoiser output variance $\|\overline{\boldsymbol{x}}_0 - \boldsymbol{x}_0\|_2^2/d$, GLM-FIRE $\overline{\sigma}_y^2$, and noise variance $\|\overline{\boldsymbol{y}} - \boldsymbol{A}\boldsymbol{x}_0\|_2^2/m$ vs. GLM-FIRE iteration for noisy CDP phase retrieval at $t[k] = 1000$ for a single validation sample $\boldsymbol{x}_0$.

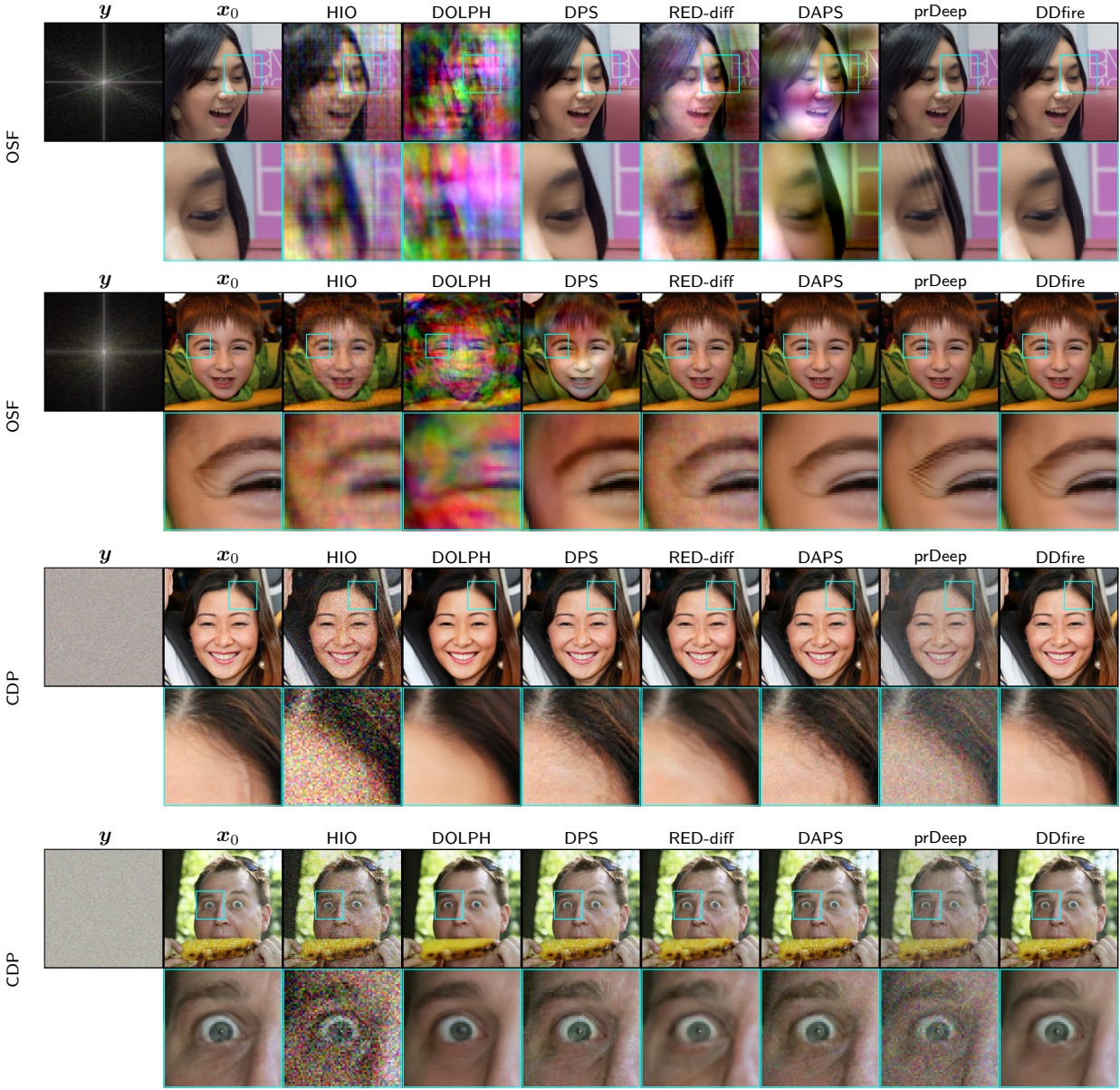

Figure 14: Example recoveries from noisy phase retrieval with FFHQ images.

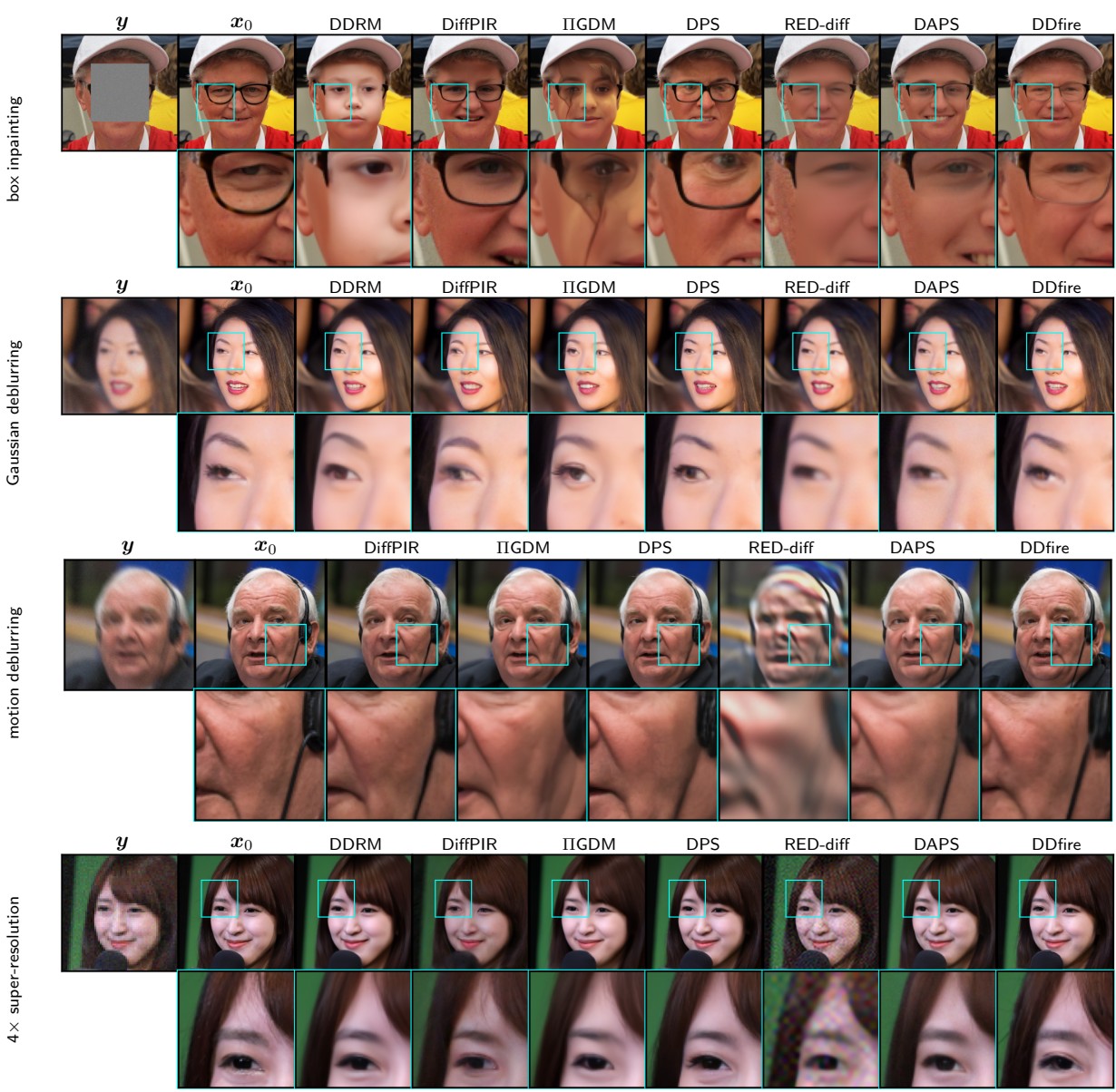

Figure 15: Example recoveries from noisy linear inverse problems with FFHQ images.

