# OpenReview forum: "Solving Inverse Problems using Diffusion with Iterative Colored Renoising"
_TMLR — Accepted by TMLR_

### Review · Reviewer_tbY8 · 2025-05-19

**Summary Of Contributions:**

This work aims to more accurately estimate the gradient of the measurement-conditional score function for solving inverse problems within the plug-and-play (PnP) framework. The authors first introduce FIRE, an iterative method that combines neural denoising with MMSE estimation and a novel colored renoising step to ensure that the denoiser input matches the white noise distribution it was trained on. FIRE is then embedded into the DDIM reverse process, resulting in the DDfire sampler. The paper provides a convergence analysis of FIRE and demonstrates, through extensive empirical results, significant improvements in reconstruction quality.

**Audience:**

Yes

**Claims And Evidence:**

No

**Requested Changes:**

- see weakness above

- in algorithm 1, should d_theta be as part of the input

**Strengths And Weaknesses:**

Strength:

- The paper provides a theoretical convergence guarantee for the FIRE algorithm and empirically demonstrates that DDfire achieves significantly improved reconstruction performance across a variety of tasks.

- Despite performing multiple inner iterations in FIRE, DDfire maintains runtime efficiency.

- The paper offers comprehensive studies for FIRE and DDfire frameworks.

Weakness and questions:
- colored re-noising step is the key contribution of the paper, it would be beneficial to include a review of related literature and empirical comparisons with other plug-and-play methods that also incorporate re-noising strategies. For example,

   M Renaud, Plug-and-play image restoration with stochastic denoising regularization

   Yuyang Hu, Stochastic Deep Restoration Priors for Imaging Inverse Problems

- how does the level of ill-posedness of A affect the reconstruction performance of both the proposed method and existing baseline approaches?

- how sensitive is DDfire to the choice of hyperparameters

---

> ### Author Response · Authors · 2025-06-12
> **Response to review (Part 1)**
>
> We thank the reviewer for the time and effort that they spent reviewing our manuscript.
> Our manuscript has been revised in accordance with their feedback and we believe that it is stronger as a result.
>
> **Weakness: It would be beneficial to include a review of related literature and empirical comparisons with other plug-and-play methods that also incorporate re-noising strategies.  For example, [Renaud'24] and [Hu'24b].**
>
> Since [Renaud'24] is based on RED from [Romano'17] and not the original plug-and-play priors framework proposed in [Venkatakrishnan'13], we interpret the reviewer's use of "plug-and-play methods" to mean any scheme that solves an inverse problem by alternating between a denoising step and a data-fidelity step.
> In that case, since Tweedie's rule says that denoising is equivalent (up to a shift and scale) to evaluating the score function of the prior, any score-based method of solving an inverse problem must also be considered a plug-and-play method.
> Under this definition, our original submission discussed the following *renoising plug-and-play* schemes:
> * DDRM (Kawar et al 2022a)
> * DDNM (Wang et al 2023)
> * DiffPIR (Zhu et al 2023)
> * PiGDM (Song et al 2023)
> * DPS (Chung et al 2023a)
> * DOLPH (Shoushtari et al 2023)
> * DDS (Chung et al 2024)
> * RED-diff (Mardani et al 2024)
> * PnP-SGS (Coeurdoux et al 2024)
> * PMC-PnP (Wu et al 2024)
> * DPnP (Xu et al 2024)
> * DAPS (Zhang et al 2025)
>
> and empirically compared our proposed approach to
> * DDRM (Kawar et al 2022a)
> * DiffPIR (Zhu et al 2023)
> * PiGDM (Song et al 2023)
> * DPS (Chung et al 2023a)
> * DOLPH (Shoushtari et al 2023)
> * DDS (Chung et al 2024)... new in the revision
> * RED-diff (Mardani et al 2024)
> * DAPS (Zhang et al 2025)
>
> because they are the standard baselines currently used by the diffusion-for-inverse-problems community.
>
> To address the reviewer's comment, we added a discussion of the following *renoising plug-and-play* schemes to the revision:
> * CSGM [Jalal'21]
> * PnP-ULA [Laumont'22]
> * GPnP [Bouman'23]
> * SNORE [Renaud'24].
>
> Furthermore, we added an Appendix G that compares FIRE/DDfire to the renoising PnP algorithms that use a prox-based data fidelity update: DDS, DiffPIR, and SNORE. The algorithms are first detailed and their differences with FIRE/DDfire are then discussed.
>
> **Additional references:**
>
> * [Venkatakrishnan'13] Venkatakrishnan SV, Bouman CA, Wohlberg B. Plug-and-play priors for model based reconstruction. GlobalSIP 2013
>
> * [Romano'17] Romano Y, Elad M, Milanfar P, The little engine that could: Regularization by denoising (RED). SIAM Journal on Imaging Sciences 2021
>
> * [Reehorst'19] Reehorst ET, Schniter P. Regularization by denoising: Clarifications and new interpretations. IEEE Transactions on Computational Imaging 2019
>
> * [Jalal'21] Jalal A, Arvinte M, Daras G, Price E, Dimakis A, Tamir J. Robust compressed sensing MRI with deep generative priors. NeurIPS 2021
>
> * [Laumont'22] Laumont R, Bortoli VD, Almansa A, Delon J, Durmus A, Pereyra M. Bayesian imaging using plug & play priors: when Langevin meets Tweedie. SIAM Journal on Imaging Sciences 2022
>
> * [Hurault'22] Hurault S, Leclaire A, Papadakis N. Gradient step denoiser for convergent plug-and-play.  ICLR 2022
>
> * [Bouman'23] Bouman CA, Buzzard GT. Generative plug and play: Posterior sampling for inverse problems. Allerton 2023
>
> * [Renaud'24] Renaud M, Prost J, Leclaire A, Papadakis N. Plug-and-play image restoration with stochastic denoising regularization. ICML 2024
>
> * [Hu'24a] Hu Y, Delbracio M, Milanfar P, Kamilov U. A restoration network as an implicit prior. ICLR 2024
>
> * [Hu'24b] Hu Y, Peng A, Gan W, Milanfar P, Delbracio M, Kamilov U. Stochastic deep restoration priors for imaging inverse problems. arXiv:2410.02057

---

> ### Author Response · Authors · 2025-06-12
> **Response to review (Part 2)**
>
> **Weakness: It would be beneficial to include a review of related literature and empirical comparisons with other plug-and-play methods that also incorporate re-noising strategies.  For example, [Renaud'24] and [Hu'24b].**
>
> We empirically evaluated the Annealed SNORE method from [Renaud'24] on box inpainting, Gaussian deblurring, motion deblurring, and 4x super resolution for FFHQ using the same denoiser as the other methods under test.
> We used $m=100$ outer iterations, 1500 total NFEs, the same $\{\sigma_i\}$ schedule as DDfire, and we tuned $\delta$, $\alpha_0$, and $\alpha_{m-1}$ to minimize LPIPS.
> All other hyperparameters were chosen as in [Renaud'24].
> The results are shown below, with DDfire included for reference.
> Overall, they show that SNORE is not competitive with DDfire or the other diffusion-based approaches evaluated in our paper.
>
> *Box Inpainting*
> | Method | PSNR | LPIPS | FID |
> | ----- | ----- | ----- | ----- |
> | SNORE | 18.18 | 0.2930 | 128.48 |
> | DDfire | **24.75** | **0.1101** | **25.26** |
>
> *Gaussian Deblurring*
> | Method | PSNR | LPIPS | FID |
> | ----- | ----- | ----- | ----- |
> | SNORE | 24.89 | 0.2777 | 87.32 |
> | DDfire | **27.10** | **0.1533** | **24.97** |
>
> *Motion Deblurring*
> | Method | PSNR | LPIPS | FID |
> | ----- | ----- | ----- | ----- |
> | SNORE | 23.42 | 0.2759 | 91.12 |
> | DDfire | **28.14** | **0.1374** | **26.12** |
>
> *4x Super Resolution*
> | Method | PSNR | LPIPS | FID |
> | ----- | ----- | ----- | ----- |
> | SNORE | 25.79 | 0.2473 | 63.79 |
> | DDfire | **27.13** | **0.1650** | **25.73** |
>
> Although the reviewer also mentions [Hu'24b], it is not actually a plug-and-play scheme in that it does not use a denoiser or score function.
> Rather, [Hu'24b] runs SNORE with a "restoration network" as proposed in [Hu'24a].
> As such, it is only distantly related to DDfire and the 16 other schemes listed above.
>
> **Question: How does the level of ill-posedness of $\mathbf{A}$ affect the reconstruction performance of both the proposed method and existing baseline approaches?**
>
> The reconstruction performance of inverse solvers does not directly depend on the level of ill-posedness of $\mathbf{A}$, since, for example, high levels of additive noise would make accurate reconstruction impossible even with a perfectly conditioned $\mathbf{A}=\mathbf{I}$.
> Rather, reconstruction performance is determined by the interaction between the prior $p(\mathbf{x})$ and the likelihood $p(\mathbf{y}|\mathbf{x},\mathbf{A})$.
> Roughly speaking, for good recovery performance, one needs that the prior is strong where the likelihood is weak.
>
> To further complicated matters, there are different ways to measure recovery performance.
> In some cases, accuracy-based metrics such as PSNR or SSIM are relevant, but in other cases perceptual metrics like LPIPS and FID may be more relevant.
> For example, in box inpainting, there is no hope to recover the true masked pixels, and so PSNR and SSIM may not be the most appropriate metrics.
> Rather, the goal of box inpainting is to hallucinate masked pixels that are perceptually consistent with the unmasked pixels.
> As a result of these various nuances, the diffusion-for-inverse-problems community tends to focus on a standard set of experiments: box inpainting, Gaussian deblurring, motion deblurring, and 4x super-resolution, with a standard noise std of 0.05, and a standard set of metrics: PSNR, SSIM, LPIPS, FID.
> This motivates the evaluation protocol adopted for our paper.
>
> **Question: How sensitive is DDfire to the choice of hyperparameters?**
>
> To evaluate the sensitivity of DDfire to the choice of hyperparameters $K$ and $\delta$, we included plots of PSNR and LPIPS across parameter grids in Appendix I of the revision.
>
> **Question: In Algorithm I, should $d_{\theta}$ be part of the input?**
>
> Yes, we added it in the revision.

---

### Review · Reviewer_RZJr · 2025-06-02

**Summary Of Contributions:**

This work addresses the inaccurate estimation of the gradient of the measurement-conditional score function. The authors propose a novel approach that iteratively re-estimates and "renoises" the estimate multiple times per diffusion step. Their method, termed Fast Iterative REnoising (FIRE), injects colored noise to ensure pre-trained diffusion models consistently encounter white noise, thereby minimizing distribution shift during inference. Theoretically, the authors prove the convergence of the FIRE algorithm. By integrating this estimation method with the DDIM reverse process, they develop DDFire, which achieves state-of-the-art performance in both accuracy and runtime across several linear inverse problems.

**Audience:**

Yes

**Claims And Evidence:**

No

**Requested Changes:**

- Method

The stochasticity in reverse diffusion models is known to play a crucial role in correcting inaccurate trajectories. However, the relative contributions of (a) correct variance computation and (b) noise injection should be more thoroughly analyzed to better understand their individual impacts.

- Theoretical Validity

The Gaussian assumption in Eq. (8) appears flawed, which calls into question the validity of subsequent derivations (Eqs. 9-14). Could the authors empirically verify any of these formulas? For instance, does the MMSE estimate in Eq. (11) hold in practice, or are there other formulas that could be more directly linked to the algorithm's performance?

- Algorithm
Regarding Eqs. (15)-(16): How should $\sigma^2 \ge \nu$ be chosen in practice? Since $\bar{x}$ is the denoising result given variance $\sigma^2$, and $\nu$ is computed using $\bar{x}$, there appears to be a circular dependency. Could the authors clarify whether my understanding is correct and explain how this is resolved?

- Approximation Error

What is the quantitative impact of the colored Gaussian noise approximation error (Eq. 17) on the final reconstruction error? A more detailed analysis of this relationship would strengthen the theoretical foundation.

- Experiments

While Figure 4 shows impressive results, how does the method perform on more challenging inverse problems? For example:
1. Different tasks beyond those shown
2. More diverse datasets
3. FIRE combined with other state-of-the-art methods

- Minor Comments

1. The abstract claims that existing approximations perform poorly early in the reverse process, but this point is not fully developed in the main text.

2. In the introduction:

How does FIRE relate to Plug-and-Play (PnP) algorithms? What implicit losses are traditionally used in such frameworks?
Many existing algorithms use pre-trained diffusion models for inverse problems - do they operate in a PnP-like manner? Clarifying these connections would better motivate FIRE's approach.

**Strengths And Weaknesses:**

- Strengths

The authors provide a rigorous theoretical analysis of flaws in existing methods, particularly focusing on how noise variance estimation can mitigate distribution shift during both training and inference.

Empirically, their method outperforms all existing approaches across varying numbers of function evaluations (NFEs), as demonstrated in Figure 4. This is particularly impressive, as most competing methods are only effective within specific NFE regimes.

- Weaknesses

The paper’s presentation does not match the quality of its results. For instance, in the theoretical section, certain concepts are introduced without clear definitions, requiring readers to infer their meaning. This ambiguity necessitates repeated reading to fully grasp the content. Additionally, there are theoretical flaws and missing points, which are detailed in the following section.

---

> ### Author Response · Authors · 2025-06-12
> **Response to review**
>
> We thank the reviewer for the time and effort that they spent reviewing our manuscript.
> Our manuscript has been revised in accordance with their feedback and we believe that it is stronger as a result.
>
> **Weakness: The relative contributions of (a) correct variance computation and (b) noise injection should be more thoroughly analyzed to better understand their individual impacts.**
>
> The fifth row of Table 1 analyzes the individual contribution of (a) correct variance computation in FIRE.
> When the denoiser-output variance $\nu$ is not estimated as in line 4 in Algorithm 1 but rather set at the precomputed average $\hat{\nu}\_{\phi}(\sigma)$ defined in equation (20), the PSNR of DDfire degrades by 1.29 dB and the LPIPS degrades from 0.1127 to 0.1755.
>
> Table 1 also shows the individual contribution of (b) FIRE's colored-noise injection, as performed in either line 9 or line 12 of Algorithm 1.
> In particular, the second row of Table 1 shows that when the noise injection is omitted entirely, the PSNR of DDfire degrades by 5.83 dB and the LPIPS degrades from 0.1127 to 0.2349.
> The third row of Table 1 shows that when the injected noise is white rather than colored, the PSNR of DDfire degrades by 0.67 and the LPIPS degrades from 0.1127 to 0.1553.
> Please note that the injected noise considered here is the one that pertains to FIRE, not the one that pertains to DDIM in line 5 of Algorithm 2.
>
> This analysis shows that both (a) and (b) have a significant impact.
>
>
>
> **Weakness:** The Gaussian assumption in Eq. (8) appears flawed, which calls into question the validity of subsequent derivations (Eqs. 9-14). Could the authors empirically verify any of these formulas?
>
> To clarify, every recovery algorithm that uses the data fidelity term in a proximal update, whether it is a diffusion algorithm (e.g., DDS, DiffPIR, DAPS) or PnP algorithm (e.g., PnP-ADMM, HQS, SNORE) or convex optimization algorithm (e.g., ADMM), uses the Gaussian assumption (8).
> Since this assumption is a standard component of many respected algorithms, we are unsure why the reviewer considers it to be flawed.
>
> Furthermore, as far as equations (9)-(14) are concerned, the Gaussianity of $\mathbf{e}$ in (8) is not needed for any of them to hold, as long as (11) is described as the "linear MMSE" estimate.
> However, referring to (8) as a "white noise" assumption would likely cause confusion among the readers, because it's standard to refer to (8) as the "Gaussian assumption".
> See, for example, Section 3.2 of the DAPS paper by Zhang et al, and Lemma 2 in the DDS paper by Chung et al.
>
> In any case, Figure 8 provides empirical evidence that (10) is accurate.
>
>
> **Weakness:** Regarding (15)-(16), how should $\sigma^2 \geq \nu$ be chosen in practice?  Since $\bar{x}$ is the denoising result given variance $\sigma^2$, and $\nu$ is computed using $\bar{x}$, there appears to be a circular dependency.
>
> The value of $\sigma^2$ used in (15)-(16) is not chosen by the user but rather automatically determined by line 6 of Algorithm 1, which reads $\sigma^2 \leftarrow \max\\{\sigma^2/\rho, \nu\\}$.
> The form of the update ensures that $\sigma^2 \geq \nu$.
>
>
> **Weakness:** What is the quantitative impact of the colored Gaussian noise approximation error (Eq. 17) on the final reconstruction error?
>
> To see the quantitative impact of the colored Gaussian noise approximation in (17), which is used only in the absence of an SVD, one can compare the first row of Table 1 (which uses the approximation) to the seventh row (which does not).
> There one sees that the PSNR is identical up to four significant digits and LPIPS degrades from 0.1127 to 0.1124.
> Thus the quantitative impact is very minor.
>
> **Weakness:** The abstract claims that existing approximations perform poorly early in the reverse process, but this point is not fully developed in the main text.
>
> In the main text, Figure 1 and the discussion after (2) demonstrate, both quantitatively and qualitatively, that existing approximations perform poorly early in the reverse process.
>
> **Weakness:** In the introduction, how does FIRE relate to Plug-and-Play (PnP) algorithms?
>
> In the Introduction, the text on page 2 explains that FIRE primarily differs from traditional PnP algorithms in that it uses colored renoising.
> Both FIRE and PnP use the prior implicit in the chosen denoiser.
> A further discussion of the difference between FIRE and traditional PnP is included in Section 3.4, "Relation to other methods", and a much more detailed discussion comparing FIRE/DDFire to the closest existing algorithms (DDS, DiffPIR, SNORE) is provided in the newly added Appendix G.

---

### Review · Reviewer_31od · 2025-06-03

**Summary Of Contributions:**

This paper introduces a novel approach aimed at improving the approximation of the posterior expectation $E[X_0|X_t, y]$ in inverse problems. The proposed method, termed FIRE, incorporates a technique referred to as colored renoising, where carefully structured colored Gaussian noise is added to the output of a linear estimator. This is done to ensure that the input to the denoiser mimics additive white Gaussian noise (AWGN), thereby reducing the distribution shift between training and inference. Two variants of FIRE are proposed: one tailored for linear inverse problems with AWGN-corrupted observations, and another for generalized models, which builds on expectation propagation (EP). Finally, the authors integrate FIRE into the DDIM framework, resulting in a new posterior sampler dubbed DDfire. Experimental results demonstrate that the proposed method achieves both high performance and computational efficiency across several benchmarks.

**Audience:**

Yes

**Broader Impact Concerns:**

There are no additional concerns related to broader societal impact that require further elaboration beyond the existing statement.

**Claims And Evidence:**

Yes

**Requested Changes:**

I would like to request the following:

- Improve the clarity and structure of Section 3.1 by: (1) Providing intuitive explanations before delving into mathematical details, (2) Clearly defining the variables at their first appearance.

- Include a more explicit comparison between DDS and FIRE: (1) Clearly articulate algorithmic differences; (2) Provide at least one quantitative benchmark comparing the two methods.

**Strengths And Weaknesses:**

**Strength**
- The algorithms are thoughtfully designed and technically sound.
- The proposed method demonstrates competitive performance and relatively fast inference compared to several benchmark methods.

**Weakness**

- Clarity of Writing: The paper’s exposition is often difficult to follow, even for readers with a strong background in the field. For instance, the phrase “possibly scaled version of $x_0$" in the second paragraph of the introduction is ambiguous and could benefit from clarification. The presentation becomes increasingly unclear in the method section. In Section 3.1, Steps S1–S4 introduce key variables such as $\nu, \rho, \hat{x}$ and $c$ without immediately defining them. These definitions are only gradually revealed across the section, which impedes readability. A more effective structure would be to begin with an intuitive overview of the method, followed by a systematic introduction of notations and their roles.

- Comparison to DDS: The proposed approach appears closely related to DDS. While the paper briefly notes that FIRE differs from DDS and DiffPIR in that $\nu$ is explicitly estimated, this distinction is not elaborated on sufficiently. A more thorough explanation of how FIRE diverges from DDS—both in terms of algorithmic structure and inference dynamics—would greatly improve clarity. Additionally, a direct empirical comparison with DDS (e.g., on one or two benchmark tasks) is warranted to contextualize the performance of FIRE.

---

> ### Author Response · Authors · 2025-06-12
> **Response to review**
>
> We thank the reviewer for the time and effort that they spent reviewing our manuscript and for the issues that they raised.
> Our manuscript has been revised in accordance and we believe that it is stronger as a result.
>
> **Weakness: Clarity of Writing: The paper’s exposition is often difficult to follow, even for readers with a strong background in the field. For instance, the phrase “possibly scaled version of" in the second paragraph of the introduction is ambiguous and could benefit from clarification.
> The presentation becomes increasingly unclear in the method section.
> In Section 3.1, Steps S1–S4 introduce key variables such as $\nu, \rho, \hat{\mathbf{x}}, \mathbf{c}$ without immediately defining them.
> These definitions are only gradually revealed across the section, which impedes readability.
> A more effective structure would be to begin with an intuitive overview of the method, followed by a systematic introduction of notations and their roles.**
>
> We have revised the Introduction and Section 3.1 in an effort to address the reviewer's concerns.
> The word "possibly" has been removed from the Introduction and an explicit link to background Section 2 is provided in case the reader wonders about "scaled version of".
> Furthermore, an intuitive overview of FIRE has been inserted before S1)-S4), and the definitions of $\nu, \rho, \hat{\mathbf{x}}, \mathbf{c}$ have been made clear.
> Please note that the S1)-S4) is itself an intuitive overview of the full FIRE approach, which is detailed in Algorithm 1.
>
> **Weakness: Comparison to DDS: The proposed approach appears closely related to DDS. While the paper briefly notes that FIRE differs from DDS and DiffPIR in that  is explicitly estimated, this distinction is not elaborated on sufficiently. A more thorough explanation of how FIRE diverges from DDS—both in terms of algorithmic structure and inference dynamics—would greatly improve clarity. Additionally, a direct empirical comparison with DDS (e.g., on one or two benchmark tasks) is warranted to contextualize the performance of FIRE.**
>
> The distinctions between DDS and FIRE/DDfire are explained briefly on pages 4 and 9.
> The revision includes a new Appendix G, which compares FIRE/DDfire to the renoising PnP algorithms that use a prox-based data fidelity update: DDS, DiffPIR, and SNORE.
> The algorithms are first detailed and their differences with FIRE/DDfire are then discussed at length.
>
> The revision also includes an empirical evaluation of DDS on eight benchmark tasks:
> FFHQ box inpainting,
> FFHQ Gaussian deblurring,
> FFHQ motion deblurring,
> FFHQ 4x super-resolution,
> ImageNet box inpainting,
> ImageNet Gaussian deblurring,
> ImageNet motion deblurring,
> and
> ImageNet 4x super-resolution.
> The results can be seen in Tables 2 and 3 on page 10.
> Although DDS performs well, it does not outperform DDfire in any metric on any task.

---

### Decision · Action_Editor_QEuE · 2025-08-20

**Recommendation:** Accept as is

**Additional Comments:**

This paper introduces FIRE, a novel posterior sampling framework for inverse problems that combines colored renoising with diffusion models. The method is technically sound, demonstrates competitive performance and efficiency, and presents a promising extension of DDS-style approaches. While the exposition could be clearer and direct comparisons to DDS should be strengthened, the contribution is original and impactful.

For the camera-ready, please revise the paper to streamline the presentation of the work.

**Audience:**

Yes

**Audience Explanation:**

Yes. The paper addresses posterior expectation approximation in inverse problems through diffusion-based methods, a topic of clear interest to both the inverse problems community and the broader machine learning audience engaged with generative modeling. The methodological novelty and practical efficiency of FIRE make the findings relevant to TMLR readers.

**Claims And Evidence:**

Yes

**Claims Explanation:**

Yes, the main claims are supported by technically sound algorithms and a reasonable set of experiments demonstrating competitive performance and efficiency. However, the evidence could be strengthened by clearer exposition in Section 3.1 and by providing direct comparisons with DDS to better contextualize the contribution.